

**Snowmobile Impacts on the Physical and Mechanical Properties of Different Snowpacks in**
**Colorado, U.S.A.**
Jared T. Heath[1,2], Steven R. Fassnacht[1,3,4,5,6*], Kevin J. Elder[7]
[1] Department of Ecosystem Science and Sustainability – Watershed Science, Colorado State
University, Fort Collins, Colorado USA 80523-1476
[2] City of Fort Collins, Water Resources & Treatment, Fort Collins, Colorado USA 80521
[3] Cooperative Institute for Research in the Atmosphere, Fort Collins, Colorado USA 80523-1375
[4] Geospatial Centroid at CSU, Fort Collins, Colorado USA 80523-1019
[5] Natural Resources Ecology Laboratory, Fort Collins, Colorado USA 80523-1499
[6] Geographisches Institut, Georg-August-Universität Göttingen, 37077 Göttingen, Germany
[7] Rocky Mountain Research Station, US Forest Service, Fort Collins, Colorado USA 80526
*Corresponding author*: steven.fassnacht@colostate.edu; phone: +1.970.491.5454
Short title: **Snowpack Changes due to Snowmobile Use**



**Abstract**
Physical and material properties of the snowpack, including snow density, temperature,
stratigraphy, hardness, and ram resistance were measured from snow pit profiles to examine the
statistical difference between no use and varying degrees of snowmobile use (low, medium and
high). The properties were examined across the entire snowpack, from the surface to its base, and
for the basal layer of the snowpack. Experimental snow compaction study plots were located
near Rabbit Ears Pass near Steamboat Springs, Colorado and at Fraser Experimental Forest near
Fraser, Colorado. Significant changes in snowpack properties are associated with snowmobile
use beginning early in the snow accumulation season when the snowpack is shallow, as well as
earlier in the winter and at the base of the snowpack. These effects were amplified when
snowmobile use occurred on a shallow snow covered environment and with increasing degrees
of snowmobile use. On the contrary, snowmobile use that began on a deeper snowpack showed
no significant changes in snowpack properties suggesting later initiation of use minimizes
impacts to snowpack properties from snowmobile use.





## 1. Introduction

Winter recreation on snow is big business; in the United States, skiing accounted for over $12 billion in 2010 (Burakowski and Magnusson, 2012) while snowmobiling accounted for between $7 billion (American Council of Snowmobile Associations, 2014) to $26 billion (International Snowmobile Manufacturers Association, 2016) annually. Across the United States, much of the snowmobile use is on public land, such as United States National Forest System with about 6 million snowmobile visits annually accessing about 327,000 km$^2$ of land (US Forest Service, 2010 and 2013a). Across the six Colorado and one southern Wyoming National Forests (NFs) there are 1.1 to 1.6 million annual snowmobile visits, with an increase from 580 thousand to 690 thousand between 2010 to 2013 in northern Colorado (Routt NF and Arapaho-Roosevelt NF) and southern Wyoming (Medicine Bow NF) (US Forest Service, 2010 and 2013a). Annually, snowmobiling added $130 million to the Colorado economy (Colorado Off-Highway Vehicle Coalition, 2016) and $125 millions to the Wyoming economy (Nagler et al., 2012). As the number of people participating in these activities increases annually (Cook and Borrie, 1995; Winter Wildlands Alliance, 2006), the presence of these human activities, especially snowmobile use, may be influencing snowpack properties in seasonally snow-covered environments. Further, as the climate changes, there will be reduced land available for snowmobiling (Tercek and Rodman, 2016), likely increasing the impact of snowmobile traffic.

There have been limited studies regarding the influence snowmobile use on snowpack properties (Keddy et al., 1979; Thumlert et al., 2013). Snowmobile use on shallow snow (10 to 20 cm deep) caused a doubling of fresh snow density, but much less impact on the underlying old snow, and had a highly significant effect upon natural vegetation below the snow (Keddy et al., 1979). For deeper snow, variation in stress on the snowpack was attributed to the type of




loading, depth and snowpack stratigraphy, stress decreased with increased depth and layer
hardness, with more cohesive or supportive layers higher in the snowpack distributing the
surface load (Thumlert et al., 2013). Most relevant studies relate to snow grooming at ski resorts
(Fahay et al., 1999; Keller et al., 2004; Spandre et al., 2016a), or to traction and mobility of
wheeled vehicles across a snowpack (Abele and Gow, 1990; Shoop et al., 2006; Pytka, 2010).
We examined the effect of snowmobile use on the physical and material properties of the
snowpack. The objectives of this research were: (1) quantify changes to physical snowpack
properties due to compaction by snowmobiles; and (2) evaluate these changes based on the
amount of use, depth of snow when snowmobile use begins, and the snowfall environment where
snowmobiles operate. This work examines both the entire snowpack and the basal layer.

**2. Study Sites**
During the 2009-2010 snow season a set of snow compaction plots were located near
Rabbit Ears Pass (REP) in the Rocky Mountains of northern Colorado to southeast of the town of
Steamboat Springs. REP is within the Medicine Bow-Routt NF (Figure 1) along the Continental
Divide encompassing over 9,400 km$^2$ (2 million acres) of land in Colorado and Wyoming.
Rabbit Ears Pass is especially popular during the winter season and is heavily used by
snowmobilers and other winter recreationalists due to the ease of access to backcountry terrain
from Colorado Highway 40. Due to heavy use and conflict among users during the winter
season, the Forest Service manages Rabbit Ears Pass for both non-motorized and motorized uses.
The west side of pass is designated for non-motorized users and prohibits the use of motorized
winter recreation and, the east side of the pass is a mixed use area and open to motorized users
(Figure 1).





Two REP experimental snow compaction study plots were located adjacent to one
another within an open meadow north of Colorado Highway 40 at an elevation of approximately
3,059 m (Figure 1). The snow compaction sites were established within an area that prohibits
motorized use to protect the study sites from unintended impacts of snowmobilers. The
Columbine snow telemetry (SNOTEL) station, located at an elevation of 2,792 m, was used to
characterize the 2009-2010 winter on REP.
Three operational sites were identified along Colorado Highway 40 on REP (Figure 1 left
inset) where the specific amount of snowmobile use was unknown. The "natural" control site
was Walton Creek, located west of Rabbit Ears Pass in an open meadow at an elevation of 2,895
m within a managed area that prohibits motorized use. Snowshoers, skiers, and snowboarders
primarily use this area in the winter to access backcountry terrain. Two treatment sites were
located east of REP at an elevation of about 2,900 m within an area managed for motorized and
mixed uses; the Dumont Lakes and Muddy Creek sites were located in open meadows near their
trailheads (Figure 1). These trailheads provide backcountry access to snowmobilers and
snowmobile use in the meadows near the trailheads is medium to high, especially on weekends
and over holidays. The meadow near the Muddy Creek trailhead is more heavily used by
snowmobiles than the meadow near the Dumont Lakes trailhead.
Another experimental snow compaction plot was established at the Fraser Experimental
Forest (FEF) near the town of Fraser, Colorado in the Rocky Mountains of Central Colorado
(Figure 1). The 93 km$^2$ experimental forest is a research unit of the United States Forest Service
(USFS) Rocky Mountain Research Station (RMRS) located within the Arapaho NF. The FEF
snow compaction site was located in a small meadow at an elevation of 2,851 m among
lodgepole pine (*Pinus contorta*) forest. The Fraser Experimental Forest is closed to snowmobile

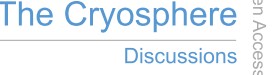

use, but is used in the winter to access backcountry terrain by snowshoers, skiers, and
snowboarders. The Berthoud Summit SNOTEL station, located at an elevation of 3,444 m, was
used to characterize the 2009-2010 winter at FEF.

**3.  Methods**
***3.1     Experimental snow compaction plots***
Snow compaction study plots were established in undisturbed areas at the REP and FEF
experimental snow compaction study areas. Each plot was 22 m wide and 15 m long. Plots were
divided into equal width transects (2 m) and treated with low, medium (FEF only), or high
snowmobile use, including a no treatment control transect representing an undisturbed
snowpack. Two control transects were used at FEF to represent the undisturbed snowpack.
Integrating two controls in the study plot allowed for replication and determination of variability.
The location of control and treatment plots across each study site was randomly selected. Each
transect was separated by a three meter buffer to eliminate the influence of compaction
treatments on adjacent transects.
Transects were treated by driving a snowmobile over the length of each transect five, 25
(FEF only) or 50 times, representing low, medium (FEF only), and high snowmobile use,
respectively. Treatments began when non-compacted snow depths were approximately 30 cm
(12 inches) for both locations, and when unpacked snow depths equaled approximately 120 cm
(48 inches) for REP only. Treatments were implemented monthly thereafter, until peak
accumulation (Figure 2). Snowpack sampling was performed within a week after each treatment,
and continued through the duration of the winter season (Figure 2).



### 3.2    Snow pit analyses and data collection


Snow pit profiles were used to examine the physical properties of the snowpack in all study sites.
A vertical snow face was excavated by digging a pit from the snow surface to the ground with
measurements of snow density, temperature, stratigraphy, hardness and ram resistance taken
vertically throughout the snowpack. Total snow depth was measured and combined with density
to yield snow water equivalent (SWE). Physical snowpack properties were compared between
non-snowmobile (control) and varying degrees (low, medium (FEF), and high) of snowmobile
use (treatment).
Density was measured at 10 cm intervals, from the surface of the snowpack to the
ground, by extracting a 250 mL or 1000 mL snow sample using a stainless steel wedge cutter
<snowmetrics.com> and measuring the mass on an electronic scale with a resolution of 1g. The
density of the snow ($\rho_s$ in kg/m$^3$) was determined by dividing the mass of the snow sample by the
volume of the wedge cutter. Snowpack density profiles and bulk snowpack density were
compared. The bulk snowpack density was determined by averaging the depth integrated density
measurements through the entire depth of the snowpack. A mean of the density measurements
for the bottom 10 cm of the snowpack were used to evaluate changes near the snow and ground
interface (basal layer).
Temperature measurements were obtained at 5 cm intervals from the top to the bottom of
the snowpack using a dial stem thermometer with $\pm 1^{\circ}$C accuracy. However, repeatability for any
given temperature is better than $\pm 1^{\circ}$C and temperature gradients are well represented by this
instrument (Elder et al., 2009; Greene et al., 2009). Snowpack temperature profiles and the
corresponding bulk temperature gradient were compared. The temperature gradient ($T_G$ in $^{\circ}$C/m)
was calculated as the ratio of the change in temperature ($\Delta T$ in $^{\circ}$C) from the point of zero

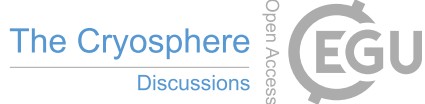



amplitude (upper boundary, 25-30 cm below the surface) and the temperature at 0 cm (lower
boundary) with the distance ($d$ in m) over which the change in temperature occurred. For this
study, the point of zero amplitude was used as the upper boundary to remove bias from diurnal
fluctuations (Pomeroy and Brun, 2001). Basal layer temperatures (0 cm) were used to compare
temperature changes near the snow and ground interface.

Stratigraphic measurements illustrate the evolution of the snowpack over time by

characterizing the shape and size of snow crystals within each stratified layer of the snowpack.
Classification of grain morphology was based on *The International Classification for Seasonal*
*Snow on the Ground* (Fierz et al., 2009) and grain size was measured and recorded to the nearest
0.5 mm using a hand lens and a crystal card. The main crystal forms / layer types were fresh,
rounded, faceted, and ice layers.

Hardness is the snowpack's compressive strength and is measured as the force per unit

area required to penetrate the structure of the snowpack (McClung and Schaerer, 2006) due to
microstructure and bonding characteristics of the snow grains (Shapiro et al., 1997). Hardness
measurements were taken horizontally with a force gauge in each stratigraphic layer using a
Wagner Instruments Force Dial gauge (<http://wagnerinstruments.com>) with maximum force
measurements of 25 N and 100 N, and fabricated circular metal plate attachments of known area.
The circular metal plate was pushed into the snow and the force required to penetrate the snow
was recorded. The snow hardness ($h_i$ in N/m$^2$) for each stratigraphic layer was calculated as the
force required to penetrate the snow ($F$ in N) per unit area of the circular metal plate ($A$ in m$^2$).
The bulk snowpack hardness ($H_B$ in N/m$^2$) was determined by weighing each stratigraphic layer
hardness measurement by the stratigraphic layer thickness. The hardness associated with the



bottom stratigraphic layer for each transect was used to describe hardness changes in the basal
layer of the snowpack.

The standard ram penetrometer is an instrument used to vertically measure the relative

hardness or resistance of a snow layers (Greene et al., 2009) and was used to assess the change in
ram resistance due to compaction through the duration of the winter season. A ram profile
measurement was taken 0.5 meters from the edge of the snow pit wall subsequent to snow pit
profile measurements. The mean ram resistance ($S_B$ in N) was determined by weighting each
stratigraphic layer's ram resistance value obtained from the standard ram penetrometer
measurement with the layer thickness. The ram resistance value associated with the bottom
stratigraphic layer was measured to describe changes in ram resistance in the basal layer of the
snowpack .

**3.3**     *Statistical analyses*

Data were analyzed using the Mann-Whitney-Wilcoxon rank sum test (Wilcoxon, 1945;

Mann and Whitney, 1947). This determines the statistical significance between two datasets,
herein different treatments compared to the control of no snowmobile use (Table 1). This
statistical test is non-parametric and determines whether two samples were selected from
populations having the same distribution. The sets samples of samples are comparable density,
temperature, hardness, and ram resistance profiles for the five different monthly measurements.
A statistical significance was determined to the 95% and 99% confidence interval ($p < 0.05$, and
$p < 0.01$) and noted with an asterisk in Table 1.

**4. Results**



The 2009-2010 winter at REP had a below average SWE based on the Columbine SNOTEL data (Figure 2). A peak SWE of 556 mm was observed on 9 April, which was 93 percent of the historical average peak SWE. Maximum snow depth measured at the REP snow compaction study plot was approximately 1.5 m and therefore represented a deep snow cover environment. From the Berthoud Summit SNOTEL data, the 2009-2010 winter at FEF had an above average SWE compared to the 29-year historical average (Figure 2). A peak SWE of 622 mm was observed on 16 May, which was 115 percent of the historical mean peak SWE. Measured snow depth at the FEF snow compaction study plot never exceeded 1 m and therefore represented a shallow snow cover environment.

### 4.1    Density

Bulk snowpack density increased at the REP snow compaction study site when low and high use compaction treatments began on 30 cm of snow (Figure 3a). As a result, low and high use compaction treatments were significantly different between these treatments (low and high) and the control, and compared to both low and high use compaction treatments beginning on 120 cm of snow (Table 1). The largest bulk snowpack density difference was observed on 6 February when the control bulk density was 246 kg/m$^3$, while the low and high use compaction treatments yielded an increase to 285 kg/m$^3$ and 328 kg/m$^3$, respectively (Figure 3a). In contrast, compaction treatments (low and high) beginning on 120 cm of snow (Figure 3b) did not significantly alter the bulk snowpack density compared to the control (Table 1). While the bulk snowpack density increased through the duration of the study period, by the last sampling date bulk snowpack density was similar between the control and treated transects (Figure 3av and 3bv). Treatment increased the density in the basal layer of the snowpack, with the largest



difference of 75% (density of 351 kg/m$^3$) and 88% (377 kg/m$^3$) for low and high use compaction
treatments observed on 12 December, respectively, compared to just over 200 kg/m$^3$ for the
control (Figure 3ai). Snow compaction treatments had little impact on basal layer densities when
treatments began on 120 cm of snow with the largest difference being observed on 6 February as
229, 234, and 268 kg/m$^3$ for the control, low and high treatments, respectively (Figure 3biii).

Bulk snowpack density also increased at the FEF snow compaction study site for all

compaction treatments (low, medium, and high use) that began on 30 cm of snow (Figure 3c).
Significant differences were observed between all treatments and the control. However, there
were no significant differences between the varying treatments (Table 1). For low and medium
use compaction treatments the largest difference in bulk snowpack density compared to the
control was on 12 February when density was measured at 177, 296, and 311 kg/m$^3$, for the
control, low and medium treatment, respectively (Figure 3ciii). Snowpack density measured for
high use had the largest difference from the control on 22 January when bulk snowpack density
was 341 kg/m$^3$ compared to a bulk density of 192 kg/m$^3$ for the control (Figure 3cii). Bulk
snowpack density generally increased during the study period, but by the end of the study period
there were minimal differences between the control and varying degrees of compaction (Figure
3cv). Basal layer density increased from all compaction treatments. After the first treatment on
27 December, the basal layer density increased by 148% (288 kg/m$^3$) for low use to about 190%
of medium and high use, compared to 116 kg/m$^3$ for the control (Figure 3ci).

### 4.2    *Temperature*

Low and high use compaction treatments at the REP snow compaction study site that began on
both a shallow snowpack of 30 cm and on a deep snowpack of 120 cm did not result in



significant changes to the temperature gradient. The maximum temperature gradients were
observed on 12 December as 18, 28, and 25$^o$C m$^{-1}$ for the control, low use, and high use
compaction treatments that began on a shallow snowpack, while they were almost the same (23,
23, and 25$^o$C m$^{-1}$) for the control, low use, and high use compaction treatments that began on a
deep snowpack. Temperature gradients for all treatments decreased throughout the winter season
until all uses exhibited a temperature gradient approaching 0$^o$C m$^{-1}$ by 17 April, favoring
sintering and bonding of snow crystals. The coldest basal layer temperatures were about -2 and -
3$^o$C on 12 December for all treatments compaction treatments began on deep and shallow
snowpack, respectively. Basal layer temperatures increased throughout the winter season until all
uses exhibited a basal layer temperature of -1$^o$C by 17 April.

Low, medium and high use compaction treatments at the FEF snow compaction study site

did not significantly impact the temperature gradient. Maximum temperature gradients for low,
medium, and high use were 30$^o$C m$^{-1}$, 13$^o$C m$^{-1}$, and 20$^o$C m$^{-1}$ on 27 December compared to 20$^o$C
m$^{-1}$ measured at the control. Temperature gradients decreased throughout the winter season until
all uses exhibited a temperature gradient near 0$^o$C m$^{-1}$ by 26 April (Figure 4b). The coldest basal
layer temperature was for medium use on 22 January (-6$^o$C), with a basal layer temperature of -
5$^o$C on 27 December for all other treatments. Basal layer temperatures increased for all uses
throughout the winter season until basal layer temperatures reached -1$^o$C by 26 April (Figure 4b).

*4.3    Hardness*
Mean snowpack hardness increased at the REP snow compaction study site following low and
high use compaction treatments that began on 30 cm of snow (Figure 5a), but only for high use
at the deeper snowpack (Figure 5b). Significant increases in hardness were observed between





treatments that began on 30 cm of snow and the control, and between compaction treatments
(low and high) that began on 120 cm of snow (Table 1). For the treatment that began on the
shallow snowpack, the maximum mean hardness for the control was 82 kPa for the control on 17
April (Figure 5av) while for the low use treatment a maximum of 174 kPa was measured on 12
December and for the high use treatment, a maximum of 487 kPa was measured on 6 February.
In contrast, mean snowpack hardness was not significantly impacted by snow compaction
treatments that began on 120 cm of snow (Table 1). Mean snowpack hardness increased
following the initial snow compaction treatments for low and high use, but subsequent
compaction treatments did not appear to have a large effect (Figure 5b and Table 1). Mean
snowpack hardness for low and high use was greater than the control following the initial snow
compaction treatment for both initiation depths (30 cm and 120 cm), but there were minimal
differences by the last sampling date (Figure 5av and 5bv).

Snow compaction treatments that began on 30 cm of snow increased basal layer hardness

(Figure 5a), but treatments that began on 120 cm of snow did not impact basal layer hardness
(Figure 5b). For the former, the maximum basal layer hardness was measured at 188 kPa (Figure
5ai) and 158 kPa (Figure 5aiii) for the low and high treatments, respectively. For both controls
and all treatments that began on 120 cm of snow (Figure 5b), the maximum basal layer hardness
was about 6 kPa.

Low, medium, and high use compaction treatments resulted in a significant increase in

mean snowpack hardness following snow compaction treatments beginning on 30 cm of snow at
the FEF snow compaction study site (Table 1). These generally increased during the study
period; however, treated transects were approaching control values by the last sampling date
(Figure 5c). For the control, the maximum mean snowpack hardness was about 25 kPa (on 26



March in Figure 5civ) while the maximum treatment hardness was orders of magnitude higher at
395 kPa (low treatment on 22 January, Figure 5cii), 780 kPa (medium treatment on 26 March,
Figure 5civ) and 4,627 kPa (high treatment on 26 March, Figure 5civ). Similarly, the maximum
basal layer hardness for the control was only 4 kPa (on 26 March, Figure 5civ) and 138, 352 and
728 kPa for low, medium and high use, respectively (Figure 5cii, 5civ, and 5civ).

### 4.4   Ram resistance
Low and high use compaction treatments at REP caused an increase in mean snowpack ram
resistance (Figure 6a and 6b), but the difference was only significant for treatments that began on
30 cm of snow (Table 1). The maximum mean snowpack ram resistance was measured as 128,
203, and 496 N for the control, low and high use, respectively (Figure 6av, 6av, and 6aiii). After
the initial snow compaction treatments mean snowpack ram resistance for low and high use was
greater than the control for the entire study period, but by the end of the study period minimal
differences were observed between treatments. Basal layer ram resistance increased as a result of
low and high use compaction treatments that began on both 30 cm (44, 614, and 1,297 N for
control, low and high use) and 120 cm of snow (44, 270 and 90 N for control, low and high use).

Snow compaction treatments at the FEF snow compaction study site caused a significant

increase in mean snowpack ram resistance (Figure 6c; Table 1). Maximum mean snowpack ram
resistance for the control was 18 N (26 March, Figure 6civ), for low and medium use it was
544N and 591N (26 March, Figure 6civ) respectively, while for high use it was measured at
866N (on 12 February, Figure 6c). Basal layer ram resistance increased following the initial
snow compaction treatments and continued to increase throughout the duration of the winter



season, with maximums of 28 (26 March), 1,220, 1,220, and 3,220 N for the control, low,
medium, and high treatments (on 12 February for all the use treatments).

*4.5     Operational Sites*
As illustrated by SWE (Figure 7d) and depth (Figure 7a), the amount of snow was similar for the
snowpits dug at the three operational sites, but not the same since they were up to 6km apart
(Figure 1). Also these were operational sites, i.e., the amount of treatment was not controlled and
was based solely on permitted use. Patterns of increased density (Figure 7a), hardness (Figure
7b) and ram resistance (Figure 7c) were similar to the previous presented experiments (Figures 3,
5, and 6) with the non-snowmobile snowpits being less dense (Figure 7a) and having layers that
were less hard (Figure 7b). For visual inspection, Muddy Creek had the most snowmobile use
and thus had the highest density throughout the winter, and the hardest snowpack for mid-winter
(Figure 7bii to 7biv) but at times was similar to Dumont Lakes.

**5. Discussion**

At rest, a snowmobile and its rider exert 4 to 10 kPa of pressure to the underlying

snowpack (assuming a track length from 0.9 to 1.4 m, width of 0.50 m, a snowmobile weight of
200 to 350 kg, and a rider weight of about 100 kg, data from
<http://www.polarisindustries.com>). This increase by less than an order of magnitude due to
snowmobile movement (Thumlert et al., 2013 measured stresses of about 10 to 20 kPa at a depth
of 30 cm below the surface of a deep snowpack). In comparison, fresh snow with a density of
100 kg/m$^3$ exerts a pressure of 0.003 kPa to the underlying snowpack (Moynier, 2006).
Snowpack loading by wheeled vehicles on a shallow snowpack was much greater, peaking at



about 350 kPa (Pytka, 2010). Grooming vehicles added a load similar to snowmobiles (Pytka,
2010), due to the larger track size. Thus, the snowpack results shown herein are transferrable to
grooming machinery.

The snowpack is persistently changing, once snow starts to accumulate on the ground.

The density of snow varies over space, time and with depth. For fresh snow, density ranges from
40 to 200 kg/m$^3$ (Diamond and Lowry, 1953; Schmidt and Gluns, 1991; Fassnacht and Soulis,
2002). The density of fresh snow can double with just one pass of a snowmobile on a very
shallow snowpack (Keddy et al., 1979), and even with more accumulation, density will increase,
but the underlying snow also gets more dense (Figures 3 and 7a).

Once snow accumulates on the ground, the meteorology alters the physical and material

properties of the snowpack from the surface down, such as changing its density and hardness.
Since the base of the snowpack remains at approximately 0$^o$C due to warm summer temperatures
and geothermal heating (Auerbach and Halfpenny, 1991; Pomeroy and Brun, 2001), variable
atmospheric air temperatures fluctuate between the relatively warm days and relatively cold
nights (McClung and Schaerer, 2006) and generate strong temperature and vapour pressure
gradients causing kinetic growth metamorphism that creates cohesionless facetted snow grains.
Conversely equilibrium metamorphism creates rounded grains that can easily sinter
(Sommerfeld, 1970; Colbeck, 1982; Colbeck, 1983; Colbeck, 1987). Rounding increases density
and snowpack strength. This increase in density and hardness is greatest compared to an
untreated snowpack in early to mid-season (January) for a deeper snowpack (REP in Figures 3a,
and 5a), and later into the snow season for the shallower snowpack (FEF in Figures 3c, and 5c).
Similar differences were found due to ski run grooming in an Australia snowpack with a 400%
increase in hardness early in the snow season but only about a 40% increase later in the winter



(Fahey et al., 1999). Snow grooming increased the average density by up to 36% compared to
non-groomed ski slopes (Fahey et al., 1999, Rixen et al., 2001).
Compaction of the snowpack changes in density, hardness and ram resistance (Figures 3,
5, 6, and 7), and results in deformation of snowthrough alterations in the ice matrix
(bonding/grain contacts) (Shapiro et al., 1997). Since hardness depends predominantly on grain
characteristics, such as bonding and grain contacts (Shapiro et al., 1997) and decreasing grain
size results in increased density, then compaction due to snowmobile use may alter the
microstructure of the snowpack (Table 2), directly influencing these physical and mechanical
properties (Table 1). Such changes were observed for varying snowmobile use beginning on two
different snow depths (REP only in Figures 3a, 5a, 6a versus Figures 3b, 5b, 6b) and for two
different snow covered environments (Figures 3c, 5c, 6c).
Field observations prior to snowmelt have revealed maximum late season snowpack
densities ranging from 290 kg/m$^3$ to 400 kg/m$^3$ with snow densities as high as 500 kg/m$^3$ during
snowmelt (Gold, 1958; Longley, 1960), while densities of depth hoar layers prior to melt were
about 300 kg/m$^3$ (Greene et al., 2009; Sturm et al., 2010). For a deep snow cover environment
(REP), compaction treatments beginning on a shallow snowpack (30 cm) resulted in a 15% and
33% increase in density for low and high use treatments, respectively (Figure 3a), observed mid-
winter (early February), similar to maximum late season natural snowpack densities (Gold, 1958;
Longley, 1960; Giddings and LaChapelle, 1962). Density differences were greatest for a shallow
snow cover environment (FEF), with high use resulting in 78% greater density (Figure 3c).
Conversely, no significant differences in density were observed when snowmobile use began on
a deep snowpack (120 cm) (Figures 3b, Table 1).

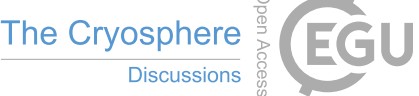



Increased densification of the snowpack due to snowmobile use influences snow hardness

(Figure 5) and ram resistance (Figure 6) due to changes in the arrangement of ice grains. In this
study, snow-hardness gauges and circular metal plates of known area were used (McClung and
Shaerer, 2006), rather than the in situ (avalanche evaluation) hand hardness test (Greene et al.,
2009). Snowmobile use beginning on a shallow snowpack (30 cm) for a deep snowpack (REP)
resulted in a 2- and 6-fold increase in maximum snow hardness for low and high use compared
to no use, whereas at a shallow snow study site (FEF), a 15-, 30- and nearly 200-fold increase in
maximum snow hardness for low, medium, and high use was observed. A shallow snow
environment is more susceptible to large changes in snow hardness due to varying snowmobile
use.

Ram resistance values ranged from 0 N to just below 1000 N, which is a normal range for

snowpack strength measurements (Colbeck et al., 1990). The precision of the ram penetrometer
used in this study was 10N so the ram resistance of an undisturbed snowpack, typically in he
range of 0.5N (Pruitt, 2005), could not be measured. These values can increase to as much as
70N as a result of two passes with one person on a snowmobile (Pruitt, 2005). Similar to
hardness observations, snowmobile use beginning on a shallow snowpack yielded ram resistance
1.5- and 4-fold greater than the natural snowpack (Figure 6). The impact of snowmobile use on a
snowpack ram resistance (Figures 6 and 7c) has only been observed by Pruitt (2005).More
frequent fresh snowfall events (REP, Figure 6a) with compaction treatments can produce a
snowpack of stratified strong and weak layers, and a deeper snowpack is capable of lessening the
effect of compaction from snowmobile use (Figure 6b).

As crystals become compacted due to snowmobile use, there is an increase in bonding

between crystals and early compaction impedes further kinetic growth. Temperature gradients



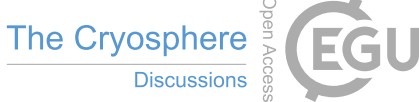

were as high as $33^o\mathrm{C\,m}^{-1}$ at the beginning of the season, about twice what was observed by de
Quervain (1958) in alpine snowpacks, and approached $0^o\mathrm{C\,m}^{-1}$ as the snowpack became isotherm
at the end of the winter season. However, temperature gradients in this study were unaffected by
compaction from snowmobile use (Figure 4, Table 1) potentially due to the edge effect of heat
transfer from the warmer ground adjacent to the plots, heat transfer from the buffer areas located
parallel to compaction transects, and diurnal changes in ambient air temperatures. The
temperature gradient was sufficient for kinetic growth metamorphism for most of the winter
season ($T_G > 10^o\mathrm{C\,m}^{-1}$), as seen by less dense lower snowpack layers for the controls (Figures 3a,
3c, 7a) and the deep snowpack where snowmobile use started at 120 cm (Figure 3b).

A decrease in crystal size was observed for both the deep and shallow snowpacks

subjected to snowmobile use (Table 2). Specifically, depth hoar crystals for the controls at FEF
reached a maximum average size of 9.0 mm, while low, medium, and high use resulted in
average crystal sizes of 1.3 mm, 2.5 mm and 1.5 mm, respectively (Table 2). While the
temperature profile differences between control and snowmobile use were not significant,
temperature gradients and thus vapour pressure gradients were less, decreasing depth hoar
growth (Table 2). Similarly, this trend was observed on REP, although the deeper snow
environment allowed growth of depth hoar but the difference in depth hoar crystal sizes between
control and treatments was less (Table 2).

The overall increase in density, hardness and ram resistance (Figure 6) was statistically

significant between the control (no snowmobile use) and all treatments, expect when treatments
were initiated on a deep snowpack (Figures 3b, 5b, and 6b, Table 1). The measured depth of
influence for a snowmobile is about 90 cm (Thumlert et al., 2013). At 20 cm below the snow
surface, the induced stress is already much less than 10 cm below the surface from a snowmobile





(Thumlert et al., 2013) or a grooming machine (Pytka, 2010). Most ski resorts in the French Alps
required a minimum snow depth of 40 cm to offer skiing, with a range from 60 cm in February to
40 cm in April (Spandre et al., 2016b). The US Forest Service (2013b) recommends a minimum
of 30 cm before the use of snowmobiles. Increasing the minimum snow depth before allowing
snowmobile traffic will reduce changes to the snowpack due to snowmobiles (Table 1).
Snowmobile use was found to have a highly significant effect upon natural vegetation
below the snow (Keddy et al., 1979), with grooming shown to delay the blooming of alpine
plants (Rixen et al., 2001) due to a later snowmelt and a significantly cooler soil (Fassnacht and
Soulis, 2002). Deeper snowpack were found to not have a cooler soil temperature under the
snowpack (Keller et al., 2004), but did melt out four weeks later, and this resulted in a cooler
snowpack at the end of the summer (Keller et al., 2004). Since the snowpack changes due to
snowmobile traffic on a shallow snowpack were significant (Table 1), the effects of snowmobile
use on the soil and vegetation underlying a shallow snowpack should be further investigated.
Snow depth will likely be less for areas with snowmobile traffic (Figure 3; Rixen et al.,
2001; Spandre et al., 2016a). However, this depends upon the meteorological conditions,
specifically the frequency and magnitude of wind. The local terrain features and position and
extent of canopy influence how the wind interacts with the snowpack (Pomeroy and Brun, 2001).
In an Australia case study, SWE increased by 45% in groomed areas (Fahey et al., 1999); at the
Rabbit Ears Pass recreational use areas, SWE also increased (Figure 7d) due to snow blowing
into the depressions created by snowmobile tracks. The increased load could further impact the
underlying snowpack properties.
Snowmaking is performed to supplement natural snow conditions. In the French Alps,
about of third of the ski slopes equipped are equipped with snowmaking facilities and this is



expected to increase, due in part to a changing climate (Spandre et al., 2016b). Artificial snow
has substantially different properties than natural snow, and adds an additional load to the
underlying snowpack (Spandre et al., 2016a). This additional snow compacts the snowpack
below it, and may create surface different conditions (Howard and Stull, 2014). Grooming of
artificial snow further compressed the snowpack (Spandre et al., 2016a). If the results presented
in this paper are extended to ski areas, the addition of artificial snow must be considered.

In Colorado alone, the economic impact of the ski industry was $4.8 billion during the

2013-14 ski season (Colorado Ski Country USA, 2015). Regardless of the use, adding mass to
the snowpack, through snowmaking (Spandre et al., 2016a), grooming (Fahey et al., 1999; Rixen
et al., 2001; Spandre et al., 2016a), or snowmobile use (Figure 7), will alter the snowpack
(Figure 3-6). A changing climate will likely reduce the extent of terrain and decrease the length
of the winter recreation season (Laxar and Williams, 2008; Steiger, 2010; Dawson and Scott,
2013; Marke et al., 2015; Tercek and Rodman, 2016). In all cases, due to climate change, more
snowmaking will be required (Steiger, 2010; Spandre et al., 2015) and this artificial snow will
impact the snowpack properties (Spandre et al., 2016a). The results presented herein are useful
when modeling the impact of grooming or snowmaking on the snowpack of ski runs (e.g.,
Howard and Stull, 2014; Marke et al., 2015; Spandre et al., 2016a).

**6. Conclusion**
This study examined the effect of compaction from snowmobile use on snowpack properties. It
showed that snowpack properties change with varying use of snowmobile use, with the amount
of snowfall, and at the initiation of use. Snowmobile use creates compaction that influences the
physical and mechanical properties of the snowpack. In particular, this increases snowpack



density, hardness, and ram resistance when winter recreational use occurs. The largest
differences in snowpack properties are associated with snowmobile use beginning on a shallow
snowpack (30 cm), which increases snowpack density, hardness, and ram resistance. These
increases are directly related to increasing snowmobile use (from low to medium to high).
Conversely, snowmobile use that begins on a deep snowpack (120 cm) has a limited effect on
snowpack properties as seen by density, temperature, hardness, and ram resistance measurements
comparable to an undisturbed snowpack.
Snowpack properties of varying snowpack environments (shallow vs. deep) respond
differently to snowmobile use. Shallow snow covers experience an increase in snowpack density,
ram resistance, and hardness that are more pronounced than changes to these properties when
snowmobile use operates on a deep snowpack. These changes in the physical properties of the
snowpack are due to snowmobile use operating on an already compacted snowpack yielding
thick layers of dense, strong, hard snow. Deep snow covers experience more snowfall events that
create "cushions" of relatively undisturbed snow between compaction events lessening the effect
of snowmobile use on snowpack properties. These differences between snow environments
suggest that shallow snowpacks are more susceptible to larger changes in snowpack properties.

**Author contribution**
The experiment were designed by J.T. Heath and S.R. Fassnacht with input from K.J. Elder. J.T.
Heath performed the experiments with assistance from K.J. Elder at the Fraser site. All authors
contributed to the writing of the manuscript.

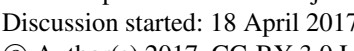

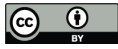

**Acknowledgments**
Appreciation goes to Robert Skorkowsky, Kent Foster and Becky Jones of the Hahns
Peak/Bears Ears Ranger District of the US Forest Service for their help and support with
compaction treatments at the Rabbit Ears Pass study site. Additional thanks goes to James
zumBrunnen of the Colorado State University Statistics Department for his assistance with
statistical interpretation. Jared Heath would also like to recognize the Colorado Mountain Club
for their help supporting this project with a generous grant. Dr. Jim Halfpenny and two
anonymous reviewers provided insight into clarifying this paper. TC editor Dr. Guillaume
Chambon provided additional comments and an important citation that helped reformulate the
discussion.

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



**Table 1.** Statistical difference (p-values) between no snowmobile use (control) and varying snow
compaction treatments on snowpack properties at the study plots located at Rabbit Ears Pass
(REP) and Fraser Experimental Forest (FEF), Colorado during the 2009-2010 winter season for
a) density, b) temperature, c) hardness, and e) ram resistance. Statistically significant differences
at the p<0.05 confident level are highlighted in grey, and highly significant (p<0.01) difference
are denoted with an asterisk.

| a) Density | | | control | Shallow initiation depth (30 cm) | | |
|---|---|---|---|---|---|---|
| | | | | Low | Medium | High |
| REP | Shallow initiation depth (30 cm) | Low | <0.01* | | | <0.01* |
| | | High | <0.01* | <0.01* | | |
| | Deep initiation depth (120 cm) | Low | 0.44 | <0.01* | | <0.01* |
| | | High | 0.24 | <0.01* | | <0.01* |
| FEF | Shallow initiation depth (30 cm) | Low | <0.01* | | 0.29 | 0.30 |
| | | Medium | <0.01* | 0.29 | | 0.98 |
| | | High | <0.01* | 0.30 | 0.98 | |


| b) Temperature | | | No use | Shallow initiation depth (30 cm) | | |
|---|---|---|---|---|---|---|
| | | | | Low | Medium | High |
| REP | Shallow initiation depth (30 cm) | Low | 0.22 | | | 0.11 |
| | | High | 0.70 | 0.11 | | |
| | Deep initiation depth (120 cm) | Low | 0.77 | 0.34 | | 0.50 |
| | | High | 1.00 | 0.22 | | 0.70 |
| FEF | Shallow initiation depth (30 cm) | Low | 0.12 | | 0.89 | 0.10 |
| | | Medium | 0.14 | 0.89 | | 0.13 |
| | | High | 0.64 | 0.10 | 0.13 | |


| c) Hardness | | | No use | Shallow initiation depth (30 cm) | | |
|---|---|---|---|---|---|---|
| | | | | Low | Medium | High |
| REP | Shallow initiation depth (30 cm) | Low | <0.01* | | | 0.16 |
| | | High | <0.01* | 0.16 | | |
| | Deep initiation depth (120 cm) | Low | 0.42 | <0.01* | | <0.01* |
| | | High | 0.06 | 0.02 | | <0.01* |
| FEF | Shallow initiation depth (30 cm) | Low | <0.01* | | 0.36 | 0.01 |
| | | Medium | <0.01* | 0.36 | | 0.08 |
| | | High | <0.01* | 0.01 | 0.08 | |


| d) Ram resistance | | | No use | Shallow initiation depth (30 cm) | | |
|---|---|---|---|---|---|---|
| | | | | Low | Medium | High |
| REP | Shallow initiation depth (30 cm) | Low | <0.01* | | | 0.08 |
| | | High | <0.01* | 0.08 | | |
| | Deep initiation depth (120 cm) | Low | 0.32 | <0.01* | | <0.01* |
| | | High | 0.07 | 0.01 | | <0.01* |
| FEF | Shallow initiation depth (30 cm) | Low | <0.01* | | 0.33 | <0.01* |
| | | Medium | <0.01* | 0.33 | | <0.01* |
| | | High | <0.01* | <0.01* | <0.01* | |




**Table 2.** Depth hoar grain size at the snow compaction study plots located at Rabbit Ears Pass (REP) and Fraser Experimental Forest (FEF), Colorado during the 2009-2010 winter season.

| | | date | Basal layer grain size [mm] | | | |
|---|---|---|---|---|---|---|
| | | | control | Low | Medium | High |
| REP | Shallow initiation depth (30 cm) | 12/12/2009 | 3.0 | 1.0 | | <0.5 |
| | | 01/09/2010 | 2.0 | 3.0 | | 1.0 |
| | | 02/06/2010 | 3.0 | 1.5 | | 1.0 |
| | | 03/13/2010 | 3.0 | 3.0 | | 1.0 |
| | | 04/17/2010 | 1.5 | 1.5 | | 1.0 |
| | Deep initiation depth (120 cm) | 12/12/2009 | 3.0 | 3.0 | | 3.0 |
| | | 01/09/2010 | 2.0 | 3.0 | | 1.5 |
| | | 02/06/2010 | 3.0 | 3.5 | | 3.0 |
| | | 03/13/2010 | 3.0 | 3.0 | | 3.5 |
| | | 04/17/2010 | 1.5 | 1.5 | | 1.5 |
| FEF | Shallow initiation depth (30 cm) | 12/27/2009 | 4.0 | 3.0 | 1.0 | 1.0 |
| | | 01/22/2010 | 3.0 | 1.0 | 2.0 | 1.5 |
| | | 02/12/2010 | 4.5 | 2.0 | 2.0 | 1.5 |
| | | 03/26/2010 | 9.0 | 1.0 | 2.5 | 1.5 |
| | | 04/26/2010 | 5.0 | 1.5 | 3.0 | 3.0 |





**List of Figures**





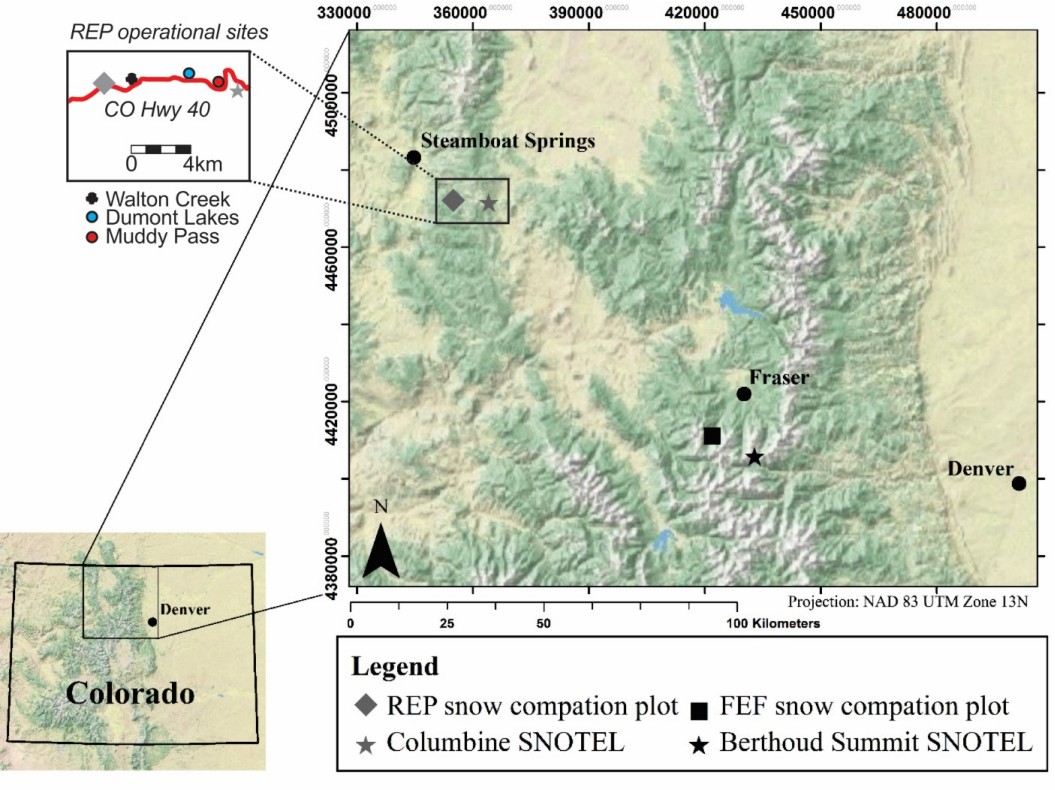

**Figure 1.** The snow compaction study plots are located near Rabbit Ears Pass in Routt National Forest and Fraser Experimental Forest in the Arapaho-Roosevelt National Forest, Colorado.





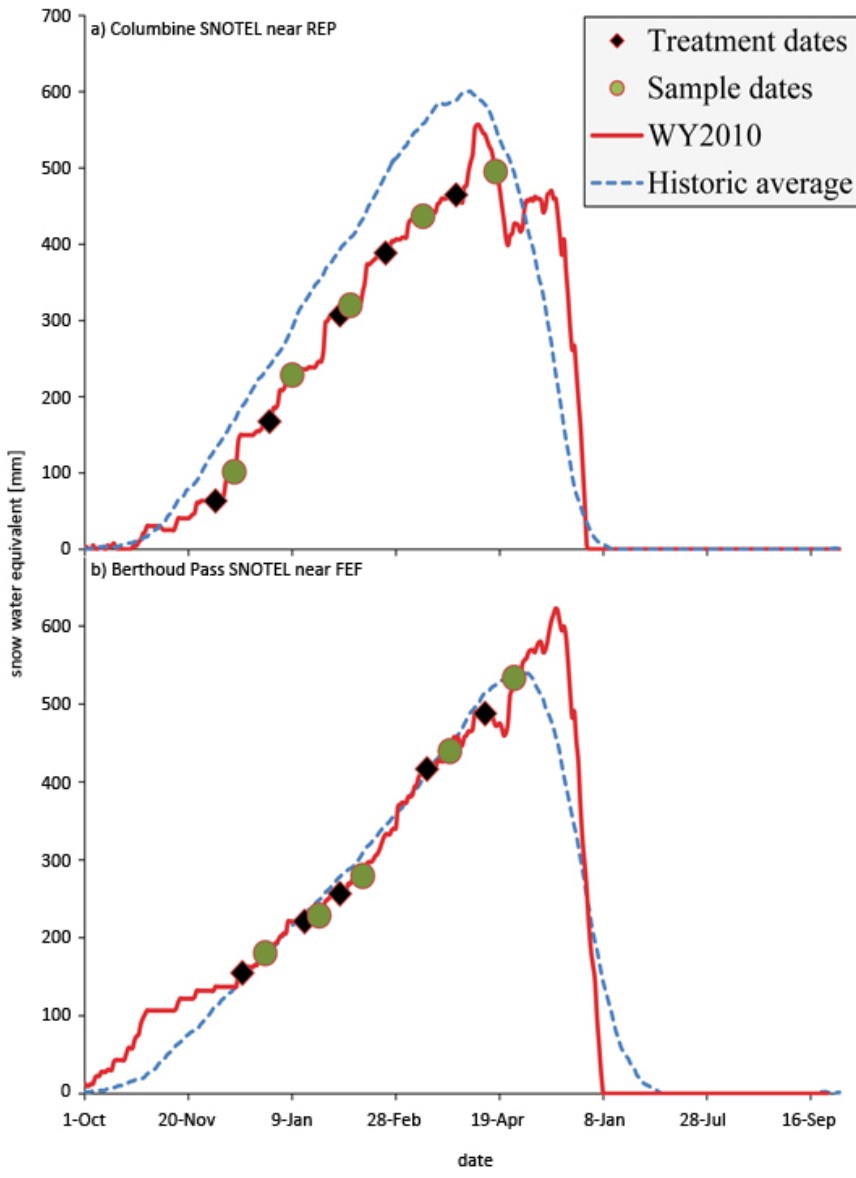

**Figure 2.** Snow water equivalent for the 2010 water year (WY2010) measured at a) the Columbine SNOTEL site near Rabbit Ears Pass, Colorado and b) the Berthoud Summit SNOTEL near Fraser Experimental Forest. Data was obtained online from the Natural Resource Conservation Service (NRCS) National Water and Climate Center (http://www.wcc.nrcs.usda.gov/).





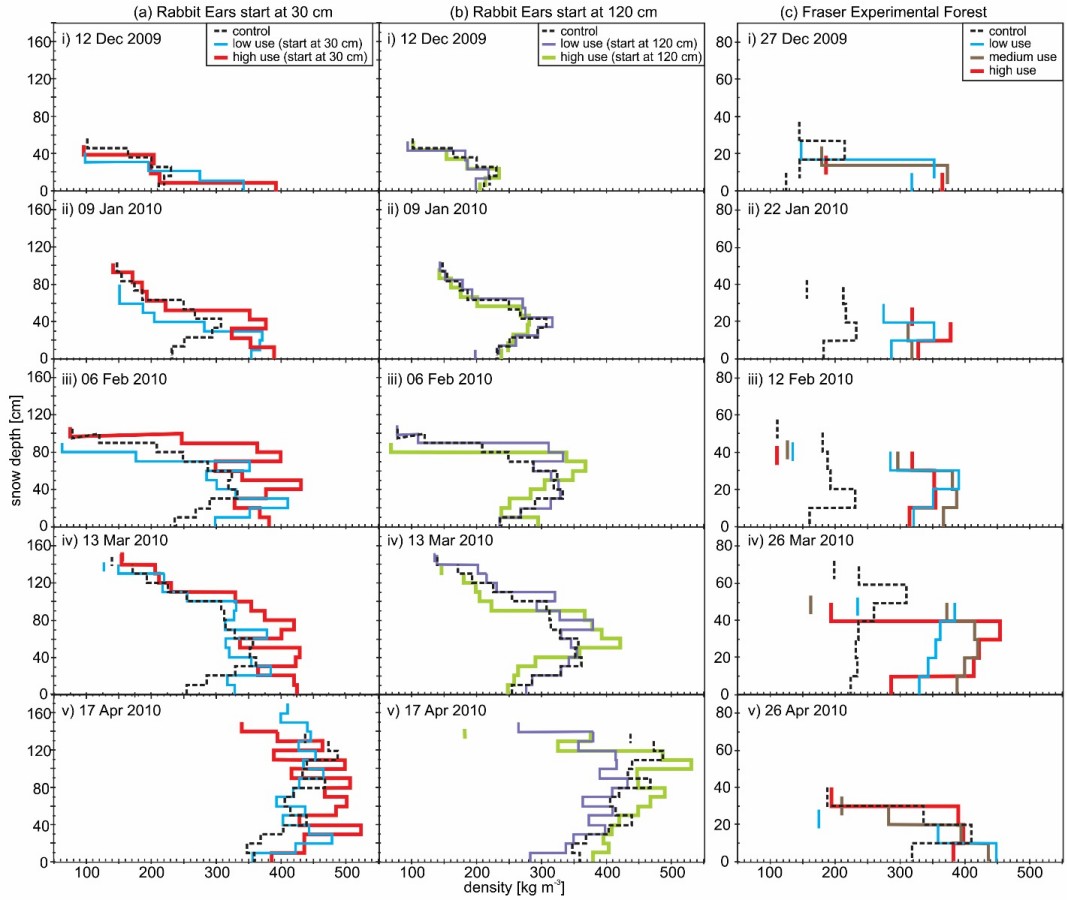

**Figure 3.** Density profiles for five dates (i to v) measured at the REP snow compaction study plot for no (control), low, and high use treatments beginning on a) 30 cm and b) 120 cm of snow, and c) the FEF snow compaction study plot for no (control), low, medium, and high use treatments beginning on 30 cm of snow. Note that free floating measurements represent overlapping density measurements.




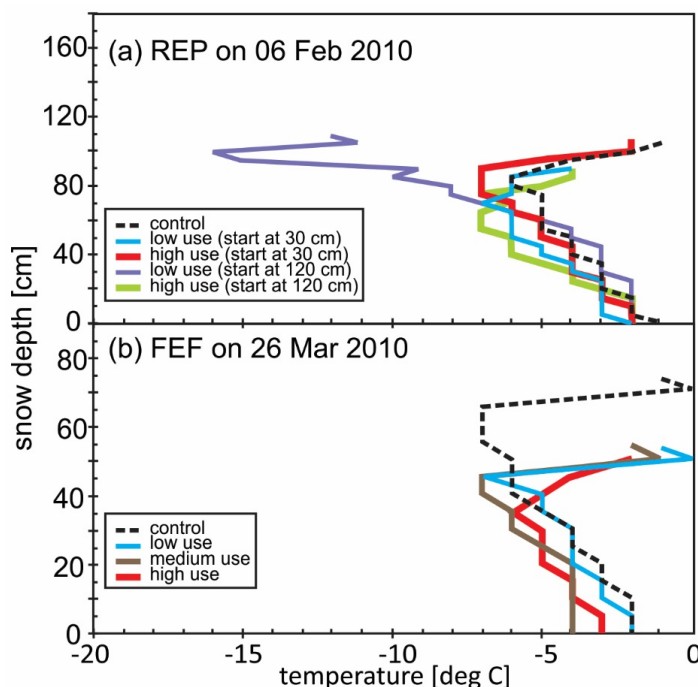

**Figure 4.** Temperature profiles measured at a) the REP snow compaction study plot on February 06, 2010 for no, low, and high use treatments beginning on 30 cm and 120 cm of snow and b) the FEF snow compaction study plot on March 26, 2010 for no, low, medium, and high use treatments beginning on 30 cm of snow.




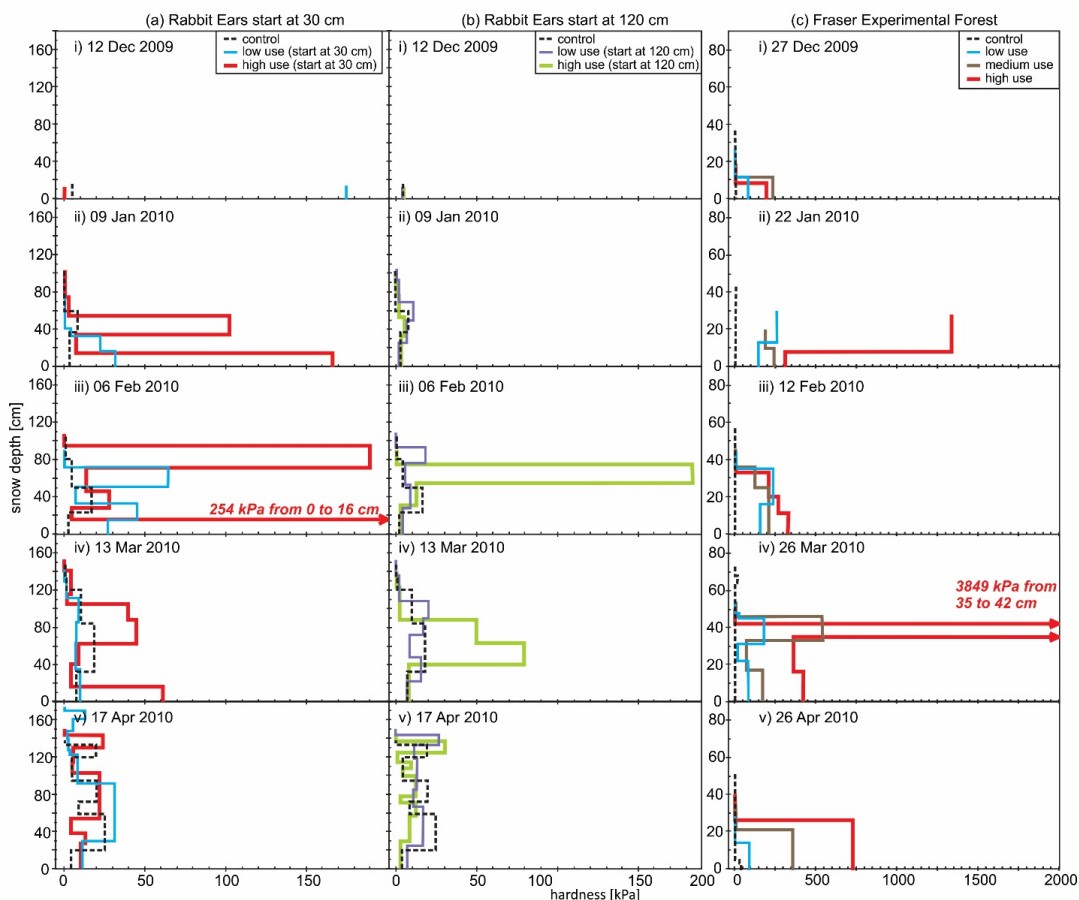

**Figure 5.** Hardness profiles for five dates (i to v) measured at the REP snow compaction study plot for no, low, and high use treatments beginning on a) 30 cm and b) 120 cm of snow, and c) the FEF snow compaction study plot for no, low, medium, and high use treatments beginning on 30 cm of snow.



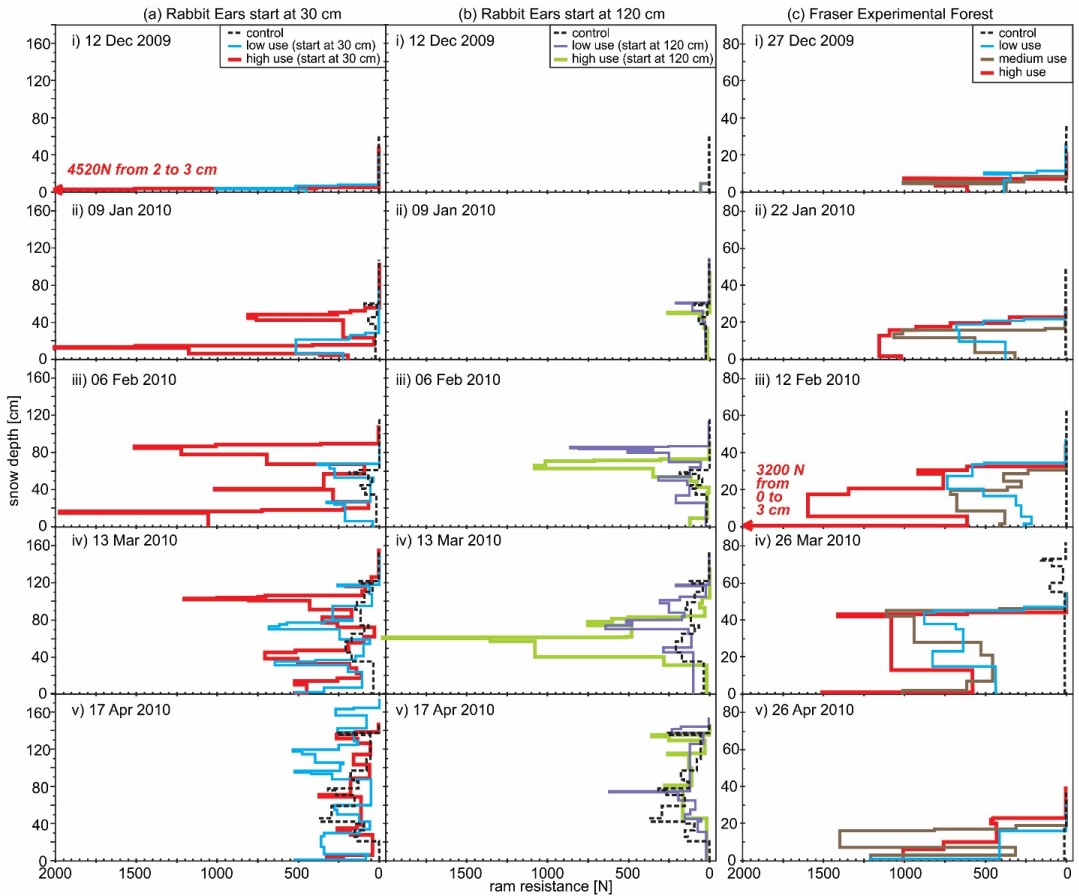

**Figure 6.** Ram resistance for five dates (i to v) profiles measured at the REP snow compaction study plot for no, low, and high use treatments beginning on a) 30 cm and b) 120 cm of snow, and c) the FEF snow compaction study plot for no, low, medium, and high use treatments beginning on 30 cm of snow.





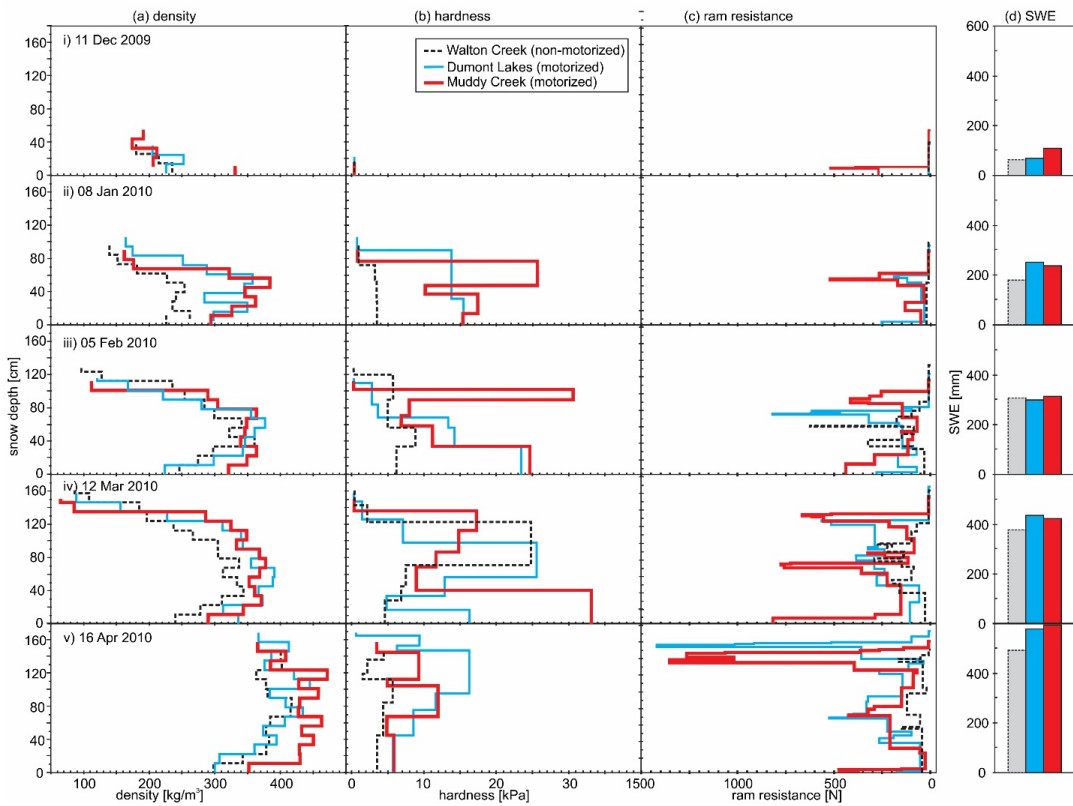

**Figure 7.** Snowpit data for Walton Creek (no snowmobile use), Dumont Lakes (moderate snowmobile use) and Muddy Creek (high snowmobile use) in the Rabbit Ears Pass recreational use areas illustrating a) density, b) hardness, c) ram resistance, and d) SWE.