# Peer review of "Snowmobile Impacts on the Physical and Mechanical Properties of Different Snowpacks in"

_The Cryosphere, 2017_

## Referee Comment (RC1) · Anonymous Referee #1 · 8 May 2017

The paper describes snow cover measurements to quantify the impact of snow mobile travel. Specifically, differences in density, hardness and temperature between an undisturbed snow cover and a snow cover subjected to various degrees of snowmobile usage are presented. The authors describe partly novel and thorough field experiments which were used to investigate these changes in detail. However, the results remain very qualitative and not very new. Furthermore, since the goal of the study is not clearly defined in the introduction and the presentation and discussion of the results is rather poor, major revisions are required before the paper can be accepted for publication.

Overall, there are three main issues with the paper:

1. After reading the introduction it does not become clear why this study is needed and why changes in snow properties due to snow mobile usage should be quantified. Indeed, the first paragraph deals with the economic importance of snowmobiling. It is completely unclear how this is at all relevant to the measurements presented in this paper. The second paragraph then lists several studies before stating the goals of this study. As such, there is no clear context, no knowledge gap is identified and it remains unclear why the authors performed these measurements.

2. The presentation of the results is rather poor and the broader relevance remains unclear. In the results section the authors show vertical density, temperature, hardness and ramm hardness profiles for all sampling dates. However, they mainly discuss mean (bulk) properties or the properties of the basal layer. As such, it would be better to show plots of the temporal evolution of the mean properties (e.g. mean density with time for the control, low use and high use) and the basal layer properties. Furthermore, the authors essentially list the results and the writing is very dry. I would suggest that the authors use the figures and tables more actively in their writing and focus on the main results. Finally, a more in-depth analysis is required to gain new insights into the effects of snow mobile travel on changes in snow cover properties and make the results more broadly relevant. Specifically, the authors could develop a simple model (e.g. linear regression) to predict snow densification after snow mobile usage and they should investigate how snow layering affects densification.

3. The discussion and conclusion sections need to be rewritten. The lack of a clear objective in the introduction translates to a very scattered discussion. Vague and out of context statements are made which do not really relate to the work presented in this paper. For instance, the third paragraph of the discussion deals with snow metamorphism. Some very general statement on the influence of ground and air temperature are made and then related to very specific increases in density observed in the measurements (lines 332 to 334). The line of thought is very hard to follow. Similarly, there are vague statements about the transferability of the results to snow grooming (lines
306-316), minimum snow depth for skiing (lines 405-409) and snow making (lines 426-433) which seem completely out of context. The authors need to do a much better job at putting their results into context, discuss the limitations of their findings and highlight new insights.

Specific comments:

line 33: It is unclear to me why climate change will affect the amount of land available for snowmobiling. line 36-39: How can there be old snow below a shallow snow cover? This sentence is very unclear and should be rewritten. line 55: remove imperial units here and throughout the paper line 58-61: it is not clear to me why this section on conflicts among different user groups is relevant to the paper. line 67: The authors should describe what a SNOTEL station is and what they measure. line 68: "... was used to characterize the 2009-2010 winter on REP". Characterize is not very specific. line 69: it is unclear what is meant by operational sites. This only became clear after reading the results. line 92-100: a sketch of the experimental setup would make this description more easy to follow. line 107: remove "and continued through the duration of the winter season". line 110-113: rewrite to "Vertical snow profiles were observed to record snowpack properties including snow density, temperature, stratigraphy hardness and ramm resistance. " line 118: mL should be ml line 118: mention the thickness of the density cutters. line 119-121: remove the sentences "The density of snow .... and bulk snowpack density were compared." line 123-125: Unclear how a mean over 10 cm can be taken if the measurements are done every 10 cm. line 127-129: "However, repeatability for any ...." it is unclear what the authors want to say here. line 131: unclear what is meant by "point of zero". Do you mean the minimum temperature? line 141-142: remove sentence "The main crystal forms...." line 148: mention the area of the metal plate attachment. line 156-160: ramm and not ram. Also, better describe how ramm measurements are made. Right now it is not clear that this is a cone penetration test. Provide a reference, e.g. Gubler (1975) line 162-163: "bottom stratigraphic layer" is not defined. Do you mean basal layer as defined I layer 125? If so, consistently use TCD
basal layer. line 171: typo "sets samples of samples" line 173-174: clearly state what you define as significant and highly significant. line 177-185: The definition of a deep and shallow snowpack seems rather arbitrary since the difference in snow depth is not very large. Furthermore, I would not qualify a snow cover of 150 cm as deep. line 223 changes in temperature gradient line 228-229: remove "favoring sintering and bonding of snow crystals" as it is not relevant here. line 229-231: rewrite this sentence line 245: unclear what is meant by "the deeper snowack" line 266: unclear what "These" refers to. line 267-268: unclear what is meant by "treated transects were approaching control values by the last sampling date" since the colored hardness profiles in bottom of figure 5c were not close to the control profile. line 269: change "orders" to "one to two orders". line 309-311: rewrite to clarify line 312: change to "on the underlying snowpack" line 322: change "also gets more dense" to "increases in density" line 325: this statement does not fit well with the temperature measurements shown in Figure 4. In particular the measurements in Figure 4b show a temperature of -4 at the base of the snow cover. It is not clear what the authors want to discuss here and this entire paragraph seems out of place. line 330-331: not clear what the authors mean by "easily sinter". Rounded grain do not sinter more readily than facetted grains, as was shown in van Herwijnen and Miller (2013). line 331-332: "Rounding increases density and snowpack strength" it is not clear what the point of this statement is. line 340: typo "snowthrough" line 360: this is speculation since the authors did not make any observations of grain arrangements. line 362: not clear what is meant by "avalanche evaluation" line 370: how can the precision of the ramm penetrometer be determined?? line 371: not clear what the authors mean by "undisturbed snowpack" since the ramm penetrometer is widely used to characterize the hardness of undisturbed snowpacks throughout the world, line 382-383; unclear how the reference to de Quervain is relevant here, line 384-387: remove this since the explanation in terms of edge effect and heat transfer from the buffer areas is very speculative and not convincing. line 396: "temperature gradients and thus vapour pressure gradients were less" unclear what this statement is based on since there was no significant difference in temperature gradients and vapour

**TCD**
pressure gradients were not measured. line 397-399: this sentence is contradictory, is it similar or different? line 405-409: unclear how these minimum snow depth guidelines fit in the discussion here. line 414-415: cooler snowpack at the end of the summer? line 418: snow depth was not less for the disturbed sites in Figure 3! line 431: typo "create surface different conditions" line 432-433: It is unclear how to consider artificial snow with the present results. line 442-444: I do not understand how the results presented in this paper can help when modelling the impact of snow grooming or snow making. line 448-449: the authors did not show that the amount of snowfall influenced their results! line 453-454: this statement is incorrect since there were no significant differences between low and high snow mobile usage. Figure 1: improve the caption and describe what is shown in the figure. Figure 2: It would be better to show snow depth rather than SWE to be consistent with the other figures. Also, there is no need to show data from July to September. Finally, please show the first of each month on the x axis. Figure 3: it would be better to show the mean snow density with time. Also, the snow depth is sometimes larger for the disturbed sites than for the undisturbed site. which seems counterintuitive. Figure 4: why are there vertical jumps in the temperature profiles? Also, it would be better to show the mean temperature gradient with time. Figure 5: The results shown in this figure are odd. It is not clear to me how and why the hardness of certain layers would decrease in the second half of the season. This is also not in line with the density measurements which show an overall increase over the course of the season. And again, it would be better to show mean hardness with time. Figure 6: better to use a logarithmic x axis. Also, show mean ramm hardness with time.

Gubler, H., 1975. On the rammsonde hardness equation. IAHS Publication, 114: 110-121. van Herwijnen, A. and Miller, D.A., 2013. Experimental and numerical investigation of the sintering rate of snow. Journal of Glaciology, 59(214): 269-274.

---

## Referee Comment (RC2) · E.H. Bair (Referee) · 16 May 2017

The is a field-based study on the impacts of snowmobiles on the snowpack in several areas in Colorado USA. I've carefully read the manuscript as well as the first referees comments, which I mostly agree with. My overall is assessment is that the study may be publishable after revision based on corrections that I've included in an annotated PDF. As the authors discuss, snowmobile use in the US is sizable yet there are very few studies on how snowmobiles affect the snowpack. In fact, I also reviewed one of the only two studies cited in the manuscript [Thumlert and Jamieson, 2015] where the impacts of snowmobiles were quantitatively measured on a backcountry snowpack. Thus, there is a significant gap in the research, but the authors do not present convinc-

ing evidence that this gap is worth addressing. The authors need to motivate the study. Why study changes in stratigraphy related to snowmobiles? Who will this research benefit?

The main conclusion that I came away with from this study is that regular snowmobile use, starting with a thin (30 cm) snowpack, results in a denser and harder snowpack with smaller basal grains. That conclusion is unsurprising, in that it could likely be predicted based on a basic understanding of snow mechanics, but given the lack of study on snowmobile effects, I still suggest the results are worth publishing. However, I worry that a reader might be tempted to conclude that snowmobiles can be used to strengthen the snowpack and prevent avalanches that fail on basal facets, similar to a boot packing program [e.g. Sahn, 2010]. While this may be true for isolated small areas, I cannot see backcountry snowmobile use reducing avalanche hazard, as the tracks will never carpet a slope densely enough. The authors should consider addressing this problematic conclusion that readers may come away with.

If the authors have any questions, I encourage them to contact me at nbair@eri.ucsb.edu

Ned Bair Earth Research Institute University of California, Santa Barbara

Sahn, K. (2010), Avalanche risk reduction in the continental climate: How to implement an effective boot packing program, Proceedings of the 2010 International Snow Science Workshop, p. 296-301.

Thumlert, S., and B. Jamieson (2015), Stress measurements from common snow slope stability tests, Cold Regions Science and Technology, 110, 38-46, doi: http://dx.doi.org/10.1016/j.coldregions.2014.11.005.

Please also note the supplement to this comment: http://www.the-cryosphere-discuss.net/tc-2017-26/tc-2017-26-RC2-supplement.pdf

[Figure]

**Supplement:**

[revised manuscript text omitted]

---

## Referee Comment (RC3) · E.H. Bair (Referee) · 16 May 2017

Change your third author to "Kelly" from "Kevin" J. Elder

---

## Author Comment (AC1) · 11 Jul 2017

Somehow the third author's given name was used here, rather than preferred name. This will be changed.

---

## Author Comment (AC2) · 1 Sep 2017

We want to thank the reviewers for good insight to help clarify many of the points that we were trying to make in this paper. We have changed the SWE time series to a more appropriate depth times series (as more data are now available), added a schematic of the control and treatment plots, and added a plot summarizing the data as a time series. The Introduction and Conclusions, as well as part of the Discussion have been rewritten to clarify the text. Below is how the reviewer comments were addressed.

REVIEWER 1 > The paper describes snow cover measurements to quantify the impact of snow mobile travel. Specifically, differences in density, hardness and temperature

[Figure]

between an undisturbed snow cover and a snow cover subjected to various degrees of snowmobile usage are presented. The authors describe partly novel and thorough field experiments which were used to investigate these changes in detail. However, the results remain very qualitative and not very new.

We disagree that the Results are qualitative – See Figures 4 through 9, and Tables 1 and 2. We also disagree that the Results are not new. As Review 2 states, there is only one similar paper in the literature (Thumlert and Jamieson, 2015).

> Furthermore, since the goal of the study is not clearly defined in the introduction and the presentation and discussion of the results is rather poor, major revisions are required before the paper can be accepted for publication.

We disagree. At the end of the Introduction, we clearly state the purpose and then the objectives of the paper: "We examined the effect of snowmobile use on the physical and material properties of the snowpack. The objectives of this research were: (1) quantify changes to physical snowpack properties due to compaction by snowmobiles; and (2) evaluate these changes based on the amount of use, depth of snow when snowmobile use begins, and the snowfall environment where snowmobiles operate."

>Overall, there are three main issues with the paper: 1. After reading the introduction it does not become clear why this study is needed and why changes in snow properties due to snow mobile usage should be quantified. Indeed, the first paragraph deals with the economic importance of snowmobiling. It is completely unclear how this is at all relevant to the measurements presented in this paper. The second paragraph then lists several studies before stating the goals of this study. As such, there is no clear context, no knowledge gap is identified and it remains unclear why the authors performed these measurements.

The first paragraph has been rewritten to be more succinct and use the economic and user data to set the stage for the work. Some of the specific details have been moved to an appendix. The second paragraph has also been rewritten to set the context
and clearly state that no other papers have examined how snowmobiles influence the physical and material properties of the snowpack.

>2. The presentation of the results is rather poor and the broader relevance remains unclear. In the results section the authors show vertical density, temperature, hardness and ramm hardness profiles for all sampling dates. However, they mainly discuss mean (bulk) properties or the properties of the basal layer. As such, it would be better to show plots of the temporal evolution of the mean properties (e.g. mean density with time for the control, low use and high use) and the basal layer properties.

We have added a set of figures summarizing the temporal evolution of the mean properties.

>Furthermore, the authors essentially list the results and the writing is very dry. I would suggest that the authors use the figures and tables more actively in their writing and focus on the main results. Many parts of the text have been rewritten.

>Finally, a more in-depth analysis is required to gain new insights into the effects of snow mobile travel on changes in snow cover properties and make the results more broadly relevant. Specifically, the authors could develop a simple model (e.g. linear regression) to predict snow densification after snow mobile usage.

While this is an interesting idea, we feel that this would yield a qualitative model. As this is an interactive discussion, I am eager to hear what this could be.

> and they should investigate how snow layering affects densification. This is beyond the scope of the paper.

>3. The discussion and conclusion sections need to be rewritten. The lack of a clear objective in the introduction translates to a very scattered discussion. Vague and out of context statements are made which do not really relate to the work presented in this paper. For instance, the third paragraph of the discussion deals with snow metamorphism. Some very general statement on the influence of ground and air temperature

are made and then related to very specific increases in density observed in the measurements (lines 332 to 334). The line of thought is very hard to follow. Similarly, there are vague statements about the transferability of the results to snow grooming (lines 306-316), minimum snow depth for skiing (lines 405-409) and snow making (lines 426-433) which seem completely out of context. The authors need to do a much better job at putting their results into context, discuss the limitations of their findings and highlight new insights.

The vague statements have been removed or significantly rewritten, as per the specific comments. With changes to the Introduction, we feel the paper is put better into context.

Specific comments: >line 33: It is unclear to me why climate change will affect the amount of land available for snowmobiling. We think that this is self-evident.

> line 36-39: How can there be old snow below a shallow snow cover? This sentence is very unclear and should be rewritten. We don't understand why this sentence is confusing. However, this sentence has been rewritten.

> line 55: remove imperial units here and throughout the paper Removed here and through, except imperial units are left in the section that discusses the initiation of snowmobile use (12" and 48") as those are the standard in the U.S.

>line 58-61: it is not clear to me why this section on conflicts among different user groups is relevant to the paper. This is setting the context for the study site. A sentence has been added to clarify this.

>line 67: The authors should describe what a SNOTEL station is and what they measure. A sentence and weblink have been added.

>line 68: ": : : was used to characterize the 2009-2010 winter on REP". Characterize is not very specific. This sentence has been changed. The point is to show "how the 2009-2010 winter compared to other winters."

[Figure]

>line 69: it is unclear what is meant by operational sites. This only became clear after reading the results. These are "not experimentally controlled." This has been added to the sentence.

>line 92-100: a sketch of the experimental setup would make this description more easy to follow. A figure has been added.

>line 107: remove "and continued through the duration of the winter season". Removed.

>line 110-113: rewrite to "Vertical snow profiles were observed to record snowpack properties including snow density, temperature, stratigraphy hardness and ramm resistance." We use the word "ram", rather than "ramm" throughout. This sentence has been rewritten as two sentences.

>line 118: mL should be ml Either of these version are SI, so mL is maintained.

>line 118: mention the thickness of the density cutters I am not sure what the reviewer is asking for here. We measured snow density as a continuous profile of discrete 10cm measurements.

>line 119-121: remove the sentences "The density of snow : : :. and bulk snowpack density were compared." The later part of this sentence was removed and the former part was rewritten.

>line 123-125: Unclear how a mean over 10 cm can be taken if the measurements are done every 10 cm. Yes, see line 118 above.

>line 127-129: "However, repeatability for any : : :" it is unclear what the authors want to say here. This sentence was rewritten.

>line 131: unclear what is meant by "point of zero". Do you mean the minimum temperature? This is rewritten as "the snowpack depth where the temperature gradient was linear"

[Figure]

>line 141-142: remove sentence "The main crystal forms: : :." This sentence has been rewritten.

>line 148: mention the area of the metal plate attachment. Added.

>line 156-160: ramm and not ram. We disagree. To be consistent we used "ram" throughout.

>Also, better describe how ramm measurements are made. Right now it is not clear that this is a cone penetration test. Provide a reference, e.g. Gubler (1975). Text has been added based on the following citation: American Avalanche Association: Snow, Weather and Avalanches: Observation Guidelines for Avalanche Programs in the United States (3rd ed.). Victor, ID, 104pp, 2016.

>line 162-163: "bottom stratigraphic layer" is not defined. Do you mean basal layer as defined I layer 125? If so, consistently use basal layer. Not necessarily. The bottom layer can be greater than the basal layer, which we define as the bottom 10 cm from the density and temperature measurements.

>line 171: typo "sets samples of samples" changed.

>line 173-174: clearly state what you define as significant and highly significant. Added.

>line 177-185: The definition of a deep and shallow snowpack seems rather arbitrary since the difference in snow depth is not very large. Furthermore, I would not qualify a snow cover of 150 cm as deep. We have changed Figure 2 to a plot of snow depth and chosen a different SNOTEL station that is more representative of the snowpack conditions at FEF. In Colorado a snowpack deeper than 1.5 meters is considered a deeper snowpack, and this was the assumption used in this paper. We changed the text accordingly.

>line 223 changes in temperature gradient changed

>line 228-229: remove "favoring sintering and bonding of snow crystals" as it is not relevant here. Removed.

>line 229-231: rewrite this sentence deleted

>line 245: unclear what is meant by "the deeper snowack" This is when use starts on a deep snowpack.

>line 266: unclear what "These" refers to. Changed to "hardness."

>line 267-268: unclear what is meant by "treated transects were approaching control values by the last sampling date" since the colored hardness profiles in bottom of figure 5c were not close to the control profile. By 17 April, hardness values were similar.

>line 269: change "orders" to "one to two orders". Changed

>line 309-311: rewrite to clarify rewritten

>line 312: change to "on the underlying snowpack" changed

>line 322: change "also gets more dense" to "increases in density" changed

>line 325: this statement does not fit well with the temperature measurements shown in Figure 4. In particular the measurements in Figure 4b show a temperature of -4 at the base of the snow cover. It is not clear what the authors want to discuss here and this entire paragraph seems out of place. Much of this paragraph has been deleted as it is not necessary.

>line 330-331: not clear what the authors mean by "easily sinter". Rounded grain do not sinter more readily than facetted grains, as was shown in van Herwijnen and Miller (2013). This has been deleted as it is not necessary.

>line 331-332: "Rounding increases density and snowpack strength" it is not clear what the point of this statement is. This has been deleted as it is not necessary.

>line 340: typo "snowthrough" changed

>line 360: this is speculation since the authors did not make any observations of grain arrangements. This has been deleted

>line 362: not clear what is meant by "avalanche evaluation" This is meant to imply a simpler method. The text has been changed.

>line 370: how can the precision of the ramm penetrometer be determined?? This is based on measurements and calculated forces.

>line 371: not clear what the authors mean by "undisturbed snowpack" since the ramm penetrometer is widely used to characterize the hardness of undisturbed snowpacks throughout the world. This sentence has been rewritten.

>line 382-383: unclear how the reference to de Quervain is relevant here. This has been removed.

>line 384-387: remove this since the explanation in terms of edge effect and heat transfer from the buffer areas is very speculative and not convincing. This sentence was deleted.

>line 396: "temperature gradients and thus vapour pressure gradients were less" unclear what this statement is based on since there was no significant difference in temperature gradients and vapour pressure gradients were not measured. We can infer vapour pressure gradients from temperature gradients. While there is no significant difference, they were still less and a difference hoar crystal size was seen.

>line 397-399: this sentence is contradictory, is it similar or different? This sentence was reworded.

>line 405-409: unclear how these minimum snow depth guidelines fit in the discussion here. The last sentence has been changed. This is an implication of the findings of this work.

>line 414-415: cooler snowpack at the end of the summer? This is deleted.

>line 418: snow depth was not less for the disturbed sites in Figure 3! This has been reworded.

>line 431: typo "create surface different conditions" This has been rewritten.

>line 432-433: It is unclear how to consider artificial snow with the present results. This paragraph has been deleted.

>line 442-444: I do not understand how the results presented in this paper can help when modelling the impact of snow grooming or snow making. This paragraph has been deleted, and replaced with one sentence mentioning snowmaking, as there could be cross-over implication. This is not explored herein.

>line 448-449: the authors did not show that the amount of snowfall influenced their results! The point is the difference between the two sites. The sentence has been reworded.

>line 453-454: this statement is incorrect since there were no significant differences between low and high snow mobile usage. This is compared to no use, as shown in Table 1.

>Figure 1: improve the caption and describe what is shown in the figure. More detail is provided.

>Figure 2: It would be better to show snow depth rather than SWE to be consistent with the other figures. Also, there is no need to show data from July to September. Finally, please show the first of each month on the x axis. This figure has been changed.

>Figure 3: it would be better to show the mean snow density with time. Also, the snow depth is sometimes larger for the disturbed sites than for the undisturbed site, which seems counterintuitive. A plot has been added.

>Figure 4: why are there vertical jumps in the temperature profiles? This is not known.

>Also, it would be better to show the mean temperature gradient with time. A plot has

been added.

>Figure 5: The results shown in this figure are odd. It is not clear to me how and why the hardness of certain layers would decrease in the second half of the season. This is also not in line with the density measurements which show an overall increase over the course of the season. And again, it would be better to show mean hardness with time. A plot has been added.

>Figure 6: better to use a logarithmic x axis. Also, show mean ramm hardness with time. Our intention is show the differences at multiple scales. Some of this may be lost using a logarithmic axis.

Gubler, H., 1975. On the rammsonde hardness equation. IAHS Publication, 114: 110-121. van Herwijnen, A. and Miller, D.A., 2013. Experimental and numerical investigation of the sintering rate of snow. Journal of Glaciology, 59(214): 269-274.

---

## Author Comment (AC3) · 1 Sep 2017

This is a field-based study on the impacts of snowmobiles on the snowpack in several areas in Colorado USA. I've carefully read the manuscript as well as the first referees comments, which I mostly agree with. My overall is assessment is that the study may be publishable after revision based on corrections that I've included in an annotated PDF. As the authors discuss, snowmobile use in the US is sizable yet there are very few studies on how snowmobiles affect the snowpack. In fact, I also reviewed one of the only two studies cited in the manuscript [Thumlert and Jamieson, 2015] where the impacts of snowmobiles were quantitatively measured on a backcountry snowpack.

[Figure]

Thus, there is a significant gap in the research, but the authors do not present convincing evidence that this gap is worth addressing. The authors need to motivate the study. »The introduction has been rewritten to highlight the lack of research in this area, as well as the number of recreational users that this could impact.

Why study changes in stratigraphy related to snowmobiles? Who will this research benefit? »This work will benefit managers who need to make decisions about multiuse areas that are used by snowmobilers. As there has been limited related work, this also provides more quantitative information on how snowmobile use changes the snowpack. The text has been changed accordingly.

The main conclusion that I came away with from this study is that regular snowmobile use, starting with a thin (30 cm) snowpack, results in a denser and harder snowpack with smaller basal grains. That conclusion is unsurprising, in that it could likely be predicted based on a basic understanding of snow mechanics, but given the lack of study on snowmobile effects, I still suggest the results are worth publishing. However, I worry that a reader might be tempted to conclude that snowmobiles can be used to strengthen the snowpack and prevent avalanches that fail on basal facets, similar to a boot packing program [e.g. Sahn, 2010]. While this may be true for isolated small areas, I cannot see backcountry snowmobile use reducing avalanche hazard, as the tracks will never carpet a slope densely enough. The authors should consider addressing this problematic conclusion that readers may come away with. »This is an interesting comment. This has been added to the discussion.
* * *

---

## Author Comment (AC4) · 1 Sep 2017

We want to thank the reviewers for good insight to help clarify many of the points that we were trying to make in this paper. We have changed the SWE time series to a more appropriate depth times series (as more data are now available), added a schematic of the control and treatment plots, and added a plot summarizing the data as a time series. The Introduction and Conclusions, as well as part of the Discussion have been rewritten to clarify the text. The responses as to how the reviewer comments were addressed are included after each review.

---

## Referee Report (RR1)

The authors have made substantial changes to the original paper and addressed some of the concerns raised by the reviewers. While this has somewhat improved the quality of the work, the manuscript still requires major revisions to improve the presentation of the results and the discussion of their limitations and implications.

The introduction has in part been rewritten. Nevertheless, it still does not provide a clear context to explain why this study is requires and why one should be interested in the influence of snowmobile use on snow properties. Does it affect the underlying vegetation, is it relevant for snow melt in the spring, does is stabilize the snow cover to reduce the avalanche danger? I suspect the last topic was what motivated the authors to perform these measurement. If so, this should clearly be stated in the introduction, and relevant studies which have investigated these effects should be discussed.

The presentation of the results has not improved much and still remains rather poor. Most figures show vertical density, temperature, hardness and ramm hardness profiles for all sampling dates. These figures are illustrative but not easy to interpret. Furthermore, the authors mainly discuss mean (bulk) properties or the properties of the basal layer. While the authors have now included a Figure showing the evolution of mean density, basal density, mean temperature gradient and mean hardness (Figure 8), this figure is only briefly mentioned at the end of the results section in a separate subsection (4.6). Furthermore, many of the results shown in this figure are repeatedly discussed before. For instance, in lines 208 to 220 the authors discuss changes in bulk density and constantly refer to Figure 4, which shows the vertical density profiles. While reading this passage, I found myself repeatedly looking at Figure 8, and it would be much more efficient and intuitive for the reader to show the plots of the mean and basal properties in each respective subsection.

Finally, the discussion and conclusion sections still need to be largely rewritten as it remains very scattered. Indeed, the authors need to do a much better job at putting their results into context, discuss the limitations of their methodologies and findings and highlight new insights. For instance, the hardness and ramm measurements have some peculiarities. In some pits specific layers sometimes have very high values which then disappear in subsequent pits. This is not observed in the control pits and highlight the difficulties in obtaining reliable hardness measurements in snow disturbed by snowmobiles. Such problems are not discussed at all by the authors even though they clearly highlight some of the limitations of this study. Similarly, the authors put a lot of weight on a 9 mm grain size measurement in one pit (section 4.5 and line 399 in the discussion) to discuss the influence of snow mobile travel on grain size. . I have dug many snow pits and have looked at countless layers of depth hoar in various snow climates (from coastal to continental), and have seldom seen depth hoar crystals of that size.
This particular measurement is therefore rather surprising to me and could very well be an outlier, and the authors should be more cautious with their interpretation.

Specific comments:

lines 37-38: it is not clear to me why I should be interested in changes in snow properties due to snowmobile travel. The context is missing.
lines 47-48: 'had a highly significant effect' In what way did this effect manifest itself?
line 57-58: 'land managers need to make decisions'. What kind of decisions do they need to make that this study will help improve?
line 147: 'where the temperature gradient was linear' it is not very clear what the authors mean here. The temperature gradient between two temperature measurements is always 'linear'.
line 157: 'fresh' is not an official crystal form. 'Precipitation particles' should be used.

line 168: 'for each stratigraphic layer'. I assume that for thin layers this was not possible. Please state the minimum layer thickness where these hardness measurements could be made.

line 175: what do the authors mean by 'relative hardness'?

lines 195-197: I would say that even for REP the snow depth was somewhat below average.

Section 4.1: include a figure showing the temporal evolution of the mean and basal layer density over the season (from Figure 8), as most of the discussion centers around bulk and basal layer density and not the vertical profiles.

Section 4.2: include a figure showing the temporal evolution of the temperature gradient and the basal layer temperature.

line 255: 'by April 26 (Figure 5b)': this figure only shows values for 26 March.

Section 4.3: include a figure showing the temporal evolution of the mean and basal layer hardness over the season.

Section 4.4: include a figure showing the temporal evolution of the mean and basal layer ram resistance over the season.

Section 4.5: include a figure showing the temporal evolution of grain size over the season. This is much more illustrative than a table. It also more clearly shows that the 9 mm measurement is likely an outlier, and that the most marked differences in grain size were at the FEF site and for the high use site at REP (see figures below)

[Figure]

Section 4.5: this section seems redundant as all these results were already addressed in the sections above.

line 331: 'were similar' in what way? Describe the similarities and differences more precisely.

Section 5: The discussion requires extensive rewriting to more clearly discuss some of the limitations of the employed methodology, highlight the main findings and discuss the results in context with other studies.

lines 339-345 Here you provide a general statement on observed densification and compare it with results from another study. In lines 355-361 you again discuss the observed densification more quantitatively. Clearly, these two sections should be combined.

lines 348-352: I don't think that compacting the snow with a snowmobile alters the snow microstructure, unless you are compacting new snow. What was the snow type when you first compacted the snow in December? Also, snow hardness is predominantly determined by density, and not grain characteristics.

lines 352-354: 'such changes' unclear what this refers to. Be more specific.

lines 361-365: unclear what the point is here.

lines 373-374: I don't agree with this statement. Your results show that for the FEF site there were very little differences between the amount of use as the densification and grain size changes were similar for low, medium and high use. For the REP site, on the other hand, the differences were more pronounced. This is one of the main findings of your work which should be highlighted and discussed much more clearly.

lines 384-386: provide an explanation why the effect of snowmobile travel is less for deeper snow covers. To me, this would mean that the initial impact of snowmobile travel, when the snow cover is still very shallow, is decisive.

lines 387-388: I do not believe that compaction impeded faceting. However, the resulting faceted snow is likely stronger (better bonded). Did you observe differences in grain *type* at the base of the snow cover?

lines 390-393: it is unclear to me how less dense snow at the base of the control plots indicates that more metamorphism took place. You can still have kinetic growth in denser snow.

lines 401-403: 'results may be transferable': what results do you mean?

lines 404-407: I do not follow your reasoning here. The results clearly show that there was no significant difference in temperature gradient. You can therefore not conclude that the vapor pressure gradients and depth hoar growth was slower since you did not measure those. All you can say is that the densification at the start led to a decrease in grain size throughout the season.

lines 408-424: The point of this section is not clear to me. Suggest rewriting.

lines 425-431: This is the first time where a context for the measurements is given. This should also be mentioned in the introduction, as this seems the main reason why these measurements were performed.

lines 440-453: this last section does not seem very relevant to me.

Section 6: The conclusions have to be rewritten to better highlight the main findings and their implications.

---

## Referee Report (RR2)

The authors have made substantial changes to the original manuscript, and it now reads much better. In particular, the presentation of the results has improved, which now greatly facilitates the interpretation of the results. Nevertheless, I believe some changes are still required before the manuscript can be accepted for publication.

1. The presentation of the results obtained at the so-called operational sites can be improved. Specifically, I would suggest that the authors present the results shown in Figure 6 in a format similar to Figure 4. This would make the comparison of the experimental sites with the operational sites much more easy.

2. Although the discussion section has improved, it is rather long and somewhat unstructured. The authors could use sub-sections with headers to provide a better structure and rearrange the paragraphs accordingly. I would suggest to split the discussion into three sub-section, e.g. (i) limitations of the measurement setup (ii) observed changes to snowpack properties and (iii) significance/impact of changes.

3. The authors now provide a simple model to predict densification due to snow mobile usage based on the number of passes, snow depth and bulk density. This is a nice addition to the paper as it shows that changes in snowpack density could be modelled. However, in the current model, if the number of passes is 0, there can still be a change in density. Perhaps a model of the form:
$$\Delta \rho_{s\_bulk} = A \times passes(B \times d_s + C \times \rho_{s\_bulk} + D)$$
would be better suited, as the change in density would go to zero when there were no passes. Finally, the authors should mention the potential merits of a density change model in the context of land use management in the discussion section and their density change model should also be mentioned in the conclusions.

4. The authors should discuss the role of spatial variability in their results in a more quantitative manner. Indeed, the authors mention that they had two control transects at FEF (lines 126-127) and that the first measurements at FEF were performed prior to any snowmobile treatment (lines 138-140). Furthermore, the deeper snow treatments at REP only started on 1 February (lines 247-249). All these data can be used to assess the typical degree of spatial variability at the experimental sites in a more quantitative manner.

Detailed comments:

line 22: change 'where there was less snow accumulation' to 'for thinner snow accumulations'
line 48: on **very** shallow snow. Also, move the reference Keddy et al in line 50 to the end of line 48
line 49: not clear how there can be an impact on the 'underlying old snow' if the snowpack is only 10 to 20 cm deep.
line 51: define what is meant by 'deeper snow cover'.
line 60: not sure what is meant by 'greater heat loss from the snowpack and underlying soil'. Does this mean that there is more cooling in the snowpack and soil? This wouldn't make sense to me.
lines 68-69: I don't see how 'and billions of dollars are spent each year on snowmobiling' is relevant here and suggest removing it
line 83: side of **the** pass … non-motorized **users**
line 125: Two control transects: these are not shown in Fig. 2b.
lines 154-155: remove this sentence
line 167: not sure what the 'point of zero amplitude' is
lines 172-173: remove 'each stratified layer of'

line 179: rewrite as 'It is due to' does not make sense to me. Hardness is not due to something, it is a property related to something

line 184-186: Did you perform multiple measurements in thicker layers and average those values, or did you only perform one hardness measurement?

line 191: replace 'tube' with 'rod'

line 192: replace 'of known weight' with 'of defined weight'

line 197: here you mention 'stratigraphic layers'. Are these the same as those identified in the manual snow profile? Usually, the layers identified in the ram profile do not correspond one-to-one to layers in the manual profile.

line 204: The statement 'This determines the statistical significance between two datasets' is inaccurate. The Mann-Whitney test is used to compare two distributions and determine if these are statistically different from each other without assuming normal distributions.

lines 224-225: rewrite to: … REP had slightly below average snow depth compared to the 15 year mean based on the Columbine SNOTEL data…

line 226: rewrite to: …9 April was at 93% if the historical …

line 229: rewrite to: …FEF was also below the 15 year …

line 237: mention that this refers to the first data point in Figure 4ii

line 283: (Table 1**c**)

line 296-297: remove sentence: 'These results are also …'

line 307: move (Table 1) to end of sentence

line 317: fragmented facetted crystals is not an official crystal type mentioned in Fierz et al. (2009)

lines 322 to 324: rewrite to … and snow depth (Figure 6a), the amount of snow was comparable for the … sites, even though they were up to …

line 326: the statement 'were similar' cannot be concluded based on what is shown in Fig. 6. Suggest showing the results as in Fig. 4

lines 334-336: The line of reasoning does not make sense to me. Just because data do not fit the expected trend, does not mean they should be excluded. It is better to argument that you want to focus on dry snow conditions, and therefore exclude the data from later in the season.

lines 338-339: it is not clear to me what ' … were not cross-correlated' means

lines 345-349: Show the results which end up with a NSCE value of 0.71 in Fig. 7. Also, since you do not control the amount of snowmobile use at these sites, you can use the model to estimate it. It makes perfect sense that it varies throughout the season, as many factors influence the amount of use, including weather and time of year (holidays).

lines 354-357: Mention density changes in % to facilitate the comparison with literature values for grooming.

line 361: change 'densification' to 'density'

line 362: 'compaction deformed fresh snow' not sure what this means and on what observations this statement is based.

lines 378-379: Figure **3ai and 3aii**

line 382: 'spatial variability between 40 to 200 kg/m3 for fresh snow' I strongly doubt that such a variability would be observed in the experimental sites. Clearly, some quantification of the spatial variability observed at the experimental sites would be in place to provide some context.

lines 386-388: this sentence seems misplaced and should be moved to the paragraph in lines 409-415

line 387: based on Figure 4, the crystals at the end of the season were no rounded crystals, but rounded facets.

line 395: rewrite to: … property changes we observed could therefore also be …

line 411: …, it could impact weak layers that cause avalanches (Saly et al., 2016), **which are typically soft layers consisting of large facetted grains (e.g. Schweizer and Jamieson, 2003; van Herwijnen and Jamieson, 2007)**

line 415: 'Do not try … ' rephrase this to say that the effects of snow mobile use on snow stability requires more investigation.
lines 428-430: also include Marty et al. (2017); Schmucki et al. (2015)
lines 432-440: move this paragraph to the start of the discussion

Fierz, C., Armstrong, R.L., Durand , Y., Etchevers, P., Greene, E., McClung, D.M., Nishimura, K., Satyawali, P.K. and Sokratov, S.A., 2009. The International Classification for Seasonal Snow on the Ground. HP-VII Technical Documents in Hydrology, 83. UNESCO-IHP, Paris, France, 90 pp.

Marty, C., Schlögl, S., Bavay, M. and Lehning, M., 2017. How much can we save? Impact of different emission scenarios on future snow cover in the Alps. The Cryosphere, 11(1): 517-529.

Schmucki, E., Marty, C., Fierz, C. and Lehning, M., 2015. Simulations of 21st century snow response to climate change in Switzerland from a set of RCMs. International Journal of Climatology, 35(11): 3262-3273.

Schweizer, J. and Jamieson, J.B., 2003. Snowpack properties for snow profile analysis. Cold Regions Science and Technology, 37(3): 233-241.

van Herwijnen, A. and Jamieson, J.B., 2007. Snowpack properties associated with fracture initiation and propagation resulting in skier-triggered dry snow slab avalanches. Cold Regions Science and Technology, 50(1-3): 13-22.

---

## Author Response (AR2)

*We want to thank the reviewers for good insight to help clarify many of the points that we were trying to make in this paper. We have changed the SWE time series to a more appropriate depth times series (as more data are now available), added a schematic of the control and treatment plots, and added a plot summarizing the data as a time series. The Introduction and Conclusions, as well as part of the Discussion have been rewritten to clarify the text. Below is how the reviewer comments were addressed.*

**REVIEWER 1**
> The paper describes snow cover measurements to quantify the impact of snow mobile travel. Specifically, differences in density, hardness and temperature between an undisturbed snow cover and a snow cover subjected to various degrees of snowmobile usage are presented. The authors describe partly novel and thorough field experiments which were used to investigate these changes in detail. However, the results remain very qualitative and not very new.

*We disagree that the Results are qualitative – See Figures 4 through 9, and Tables 1 and 2. We also disagree that the Results are not new. As Review 2 states, there is only one similar paper in the literature (Thumlert and Jamieson, 2015).*

> Furthermore, since the goal of the study is not clearly defined in the introduction and the presentation and discussion of the results is rather poor, major revisions are required before the paper can be accepted for publication.

*We disagree. At the end of the Introduction, we clearly state the purpose and then the objectives of the paper: "We examined the effect of snowmobile use on the physical and material properties of the snowpack. The objectives of this research were: (1) quantify changes to physical snowpack properties due to compaction by snowmobiles; and (2) evaluate these changes based on the amount of use, depth of snow when snowmobile use begins, and the snowfall environment where snowmobiles operate."*

>Overall, there are three main issues with the paper:
1. After reading the introduction it does not become clear why this study is needed and why changes in snow properties due to snow mobile usage should be quantified. Indeed, the first paragraph deals with the economic importance of snowmobiling. It is completely unclear how this is at all relevant to the measurements presented in this paper. The second paragraph then lists several studies before stating the goals of this study. As such, there is no clear context, no knowledge gap is identified and it remains unclear why the authors performed these measurements.

*The first paragraph has been rewritten to be more succinct and use the economic and user data to set the stage for the work. Some of the specific details have been moved to an appendix. The second paragraph has also been rewritten to set the context and clearly state that no other papers have examined how snowmobiles influence the physical and material properties of the snowpack.*

>2. The presentation of the results is rather poor and the broader relevance remains unclear. In the results section the authors show vertical density, temperature, hardness and ramm hardness profiles for all sampling dates. However, they mainly discuss mean (bulk) properties or the properties of the basal layer. As such, it would be better to show plots of the temporal evolution of the mean properties (e.g. mean density with time for the control, low use and high use) and the basal layer properties.

*We have added a set of figures summarizing the temporal evolution of the mean properties.*

>Furthermore, the authors essentially list the results and the writing is very dry. I would suggest that the authors use the figures and tables more actively in their writing and focus on the main results.
*Many parts of the text have been rewritten.*

>Finally, a more in-depth analysis is required to gain new insights into the effects of snow mobile travel on changes in snow cover properties and make the results more broadly relevant. Specifically, the authors could develop a simple model (e.g. linear regression) to predict snow densification after snow mobile usage.

*While this is an interesting idea, we feel that this would yield a qualitative model. As this is an interactive discussion, I am eager to hear what this could be.*

> and they should investigate how snow layering affects densification.
*This is beyond the scope of the paper.*

>3. The discussion and conclusion sections need to be rewritten. The lack of a clear objective in the introduction translates to a very scattered discussion. Vague and out of context statements are made which do not really relate to the work presented in this paper. For instance, the third paragraph of the discussion deals with snow metamorphism. Some very general statement on the influence of ground and air temperature are made and then related to very specific increases in density observed in the measurements (lines 332 to 334). The line of thought is very hard to follow. Similarly, there are vague statements about the transferability of the results to snow grooming (lines 306-316), minimum snow depth for skiing (lines 405-409) and snow making (lines 426-433) which seem completely out of context. The authors need to do a much better job at putting their results into context, discuss the limitations of their findings and highlight new insights.

*The vague statements have been removed or significantly rewritten, as per the specific comments. With changes to the Introduction, we feel the paper is put better into context.*

Specific comments:
>line 33: It is unclear to me why climate change will affect the amount of land available for snowmobiling.
*We think that this is self-evident.*

> line 36-39: How can there be old snow below a shallow snow cover? This sentence is very unclear and should be rewritten.
*We don't understand why this sentence is confusing. However, this sentence has been rewritten.*

> line 55: remove imperial units here and throughout the paper
*Removed here and through, except imperial units are left in the section that discusses the initiation of snowmobile use (12" and 48") as those are the standard in the U.S.*

>line 58-61: it is not clear to me why this section on conflicts among different user groups is relevant to the paper.
*This is setting the context for the study site. A sentence has been added to clarify this.*

>line 67: The authors should describe what a SNOTEL station is and what they measure.
*A sentence and weblink have been added.*

>line 68: ": : : was used to characterize the 2009-2010 winter on REP". Characterize is not very specific.
*This sentence has been changed. The point is to show "how the 2009-2010 winter compared to other winters."*

>line 69: it is unclear what is meant by operational sites. This only became clear after reading the results.
*These are "not experimentally controlled." This has been added to the sentence.*

>line 92-100: a sketch of the experimental setup would make this description more easy to follow.
*A figure has been added.*

>line 107: remove "and continued through the duration of the winter season".
*Removed.*

>line 110-113: rewrite to "Vertical snow profiles were observed to record snowpack properties including snow density, temperature, stratigraphy hardness and ramm resistance."
*We use the word "ram", rather than "ramm" throughout. This sentence has been rewritten as two sentences.*

>line 118: mL should be ml
*Either of these version are SI, so mL is maintained.*

>line 118: mention the thickness of the density cutters
*I am not sure what the reviewer is asking for here. We measured snow density as a continuous profile of discrete 10cm measurements.*

>line 119-121: remove the sentences "The density of snow : : :. and bulk snowpack density were compared."
*The later part of this sentence was removed and the former part was rewritten.*

>line 123-125: Unclear how a mean over 10 cm can be taken if the measurements are done every 10 cm.

*Yes, see line 118 above.*

>line 127-129: "However, repeatability for any : : :" it is unclear what the authors want to say here.
*This sentence was rewritten.*

>line 131: unclear what is meant by "point of zero". Do you mean the minimum temperature?
*This is rewritten as "the snowpack depth where the temperature gradient was linear"*

>line 141-142: remove sentence "The main crystal forms: : :."
*This sentence has been rewritten.*

>line 148: mention the area of the metal plate attachment.
*Added.*

>line 156-160: ramm and not ram.
*We disagree. To be consistent we used "ram" throughout.*

>Also, better describe how ramm measurements are made. Right now it is not clear that this is a cone penetration test. Provide a reference, e.g. Gubler (1975).
*Text has been added based on the following citation:*
*American Avalanche Association: Snow, Weather and Avalanches: Observation Guidelines for Avalanche Programs in the United States (3$^{rd}$ ed.). Victor, ID, 104pp, 2016.*

>line 162-163: "bottom stratigraphic layer" is not defined. Do you mean basal layer as defined I layer 125? If so, consistently use basal layer.
*Not necessarily. The bottom layer can be greater than the basal layer, which we define as the bottom 10 cm from the density and temperature measurements.*

>line 171: typo "sets samples of samples"
*changed.*

>line 173-174: clearly state what you define as significant and highly significant.
*Added.*

>line 177-185: The definition of a deep and shallow snowpack seems rather arbitrary since the difference in snow depth is not very large. Furthermore, I would not qualify a snow cover of 150 cm as deep.
*We have changed Figure 2 to a plot of snow depth and chosen a different SNOTEL station that is more representative of the snowpack conditions at FEF. In Colorado a snowpack deeper than 1.5 meters is considered a deeper snowpack, and this was the assumption used in this paper. We changed the text accordingly.*

>line 223 changes in temperature gradient
*changed*

>line 228-229: remove "favoring sintering and bonding of snow crystals" as it is not relevant here.
*Removed.*

>line 229-231: rewrite this sentence
*deleted*

>line 245: unclear what is meant by "the deeper snowack"
*This is when use starts on a deep snowpack.*

>line 266: unclear what "These" refers to.
*Changed to "hardness."*

>line 267-268: unclear what is meant by "treated transects were approaching control values by the last sampling date" since the colored hardness profiles in bottom of figure 5c were not close to the control profile.
*By 17 April, hardness values were similar.*

>line 269: change "orders" to "one to two orders".
*Changed*

>line 309-311: rewrite to clarify
*rewritten*

>line 312: change to "on the underlying snowpack"
*changed*

>line 322: change "also gets more dense" to "increases in density"
*changed*

>line 325: this statement does not fit well with the temperature measurements shown in Figure 4. In particular the measurements in Figure 4b show a temperature of -4 at the base of the snow cover. It is not clear what the authors want to discuss here and this entire paragraph seems out of place.
*Much of this paragraph has been deleted as it is not necessary.*

>line 330-331: not clear what the authors mean by "easily sinter". Rounded grain do not sinter more readily than facetted grains, as was shown in van Herwijnen and Miller (2013).
*This has been deleted as it is not necessary.*

>line 331-332: "Rounding increases density and snowpack strength" it is not clear what the point of this statement is.
*This has been deleted as it is not necessary.*

>line 340: typo "snowthrough"
*changed*

>line 360: this is speculation since the authors did not make any observations of grain arrangements.
*This has been deleted*

>line 362: not clear what is meant by "avalanche evaluation"
*This is meant to imply a simpler method. The text has been changed.*

>line 370: how can the precision of the ramm penetrometer be determined??
*This is based on measurements and calculated forces.*

>line 371: not clear what the authors mean by "undisturbed snowpack" since the ramm penetrometer is widely used to characterize the hardness of undisturbed snowpacks throughout the world.
*This sentence has been rewritten.*

>line 382-383: unclear how the reference to de Quervain is relevant here.
*This has been removed.*

>line 384-387: remove this since the explanation in terms of edge effect and heat transfer from the buffer areas is very speculative and not convincing.
*This sentence was deleted.*

>line 396: "temperature gradients and thus vapour pressure gradients were less" unclear what this statement is based on since there was no significant difference in temperature gradients and vapour pressure gradients were not measured.
*We can infer vapour pressure gradients from temperature gradients. While there is no significant difference, they were still less and a difference hoar crystal size was seen.*

>line 397-399: this sentence is contradictory, is it similar or different?
*This sentence was reworded.*

>line 405-409: unclear how these minimum snow depth guidelines fit in the discussion here.
*The last sentence has been changed. This is an implication of the findings of this work.*

>line 414-415: cooler snowpack at the end of the summer?
*This is deleted.*

>line 418: snow depth was not less for the disturbed sites in Figure 3!
*This has been reworded.*

>line 431: typo "create surface different conditions"
*This has been rewritten.*

>line 432-433: It is unclear how to consider artificial snow with the present results.
*This paragraph has been deleted.*

>line 442-444: I do not understand how the results presented in this paper can help when modelling the impact of snow grooming or snow making.
*This paragraph has been deleted, and replaced with one sentence mentioning snowmaking, as there could be cross-over implication. This is not explored herein.*

>line 448-449: the authors did not show that the amount of snowfall influenced their results!
*The point is the difference between the two sites. The sentence has been reworded.*

>line 453-454: this statement is incorrect since there were no significant differences between low and high snow mobile usage.
*This is compared to no use, as shown in Table 1.*

>Figure 1: improve the caption and describe what is shown in the figure.
*More detail is provided.*

>Figure 2: It would be better to show snow depth rather than SWE to be consistent with the other figures. Also, there is no need to show data from July to September. Finally, please show the first of each month on the x axis.
*This figure has been changed.*

>Figure 3: it would be better to show the mean snow density with time. Also, the snow depth is sometimes larger for the disturbed sites than for the undisturbed site, which seems counterintuitive.
*A plot has been added.*

>Figure 4: why are there vertical jumps in the temperature profiles?
*This is not known.*

>Also, it would be better to show the mean temperature gradient with time.
*A plot has been added.*

>Figure 5: The results shown in this figure are odd. It is not clear to me how and why the hardness of certain layers would decrease in the second half of the season. This is also not in line with the density measurements which show an overall increase over the course of the season. And again, it would be better to show mean hardness with time.
*A plot has been added.*

>Figure 6: better to use a logarithmic x axis. Also, show mean ramm hardness with time.
*Our intention is show the differences at multiple scales. Some of this may be lost using a logarithmic axis.*

Gubler, H., 1975. On the rammsonde hardness equation. IAHS Publication, 114: 110-121.
van Herwijnen, A. and Miller, D.A., 2013. Experimental and numerical investigation of the sintering rate of snow. Journal of Glaciology, 59(214): 269-274.

**REVIEWER 2 (Edward Bair)**
This is a field-based study on the impacts of snowmobiles on the snowpack in several areas in Colorado USA. I've carefully read the manuscript as well as the first referees comments, which I mostly agree with. My overall is assessment is that the study may be publishable after revision based on corrections that I've included in an annotated PDF. As the authors discuss, snowmobile use in the US is sizable yet there are very few studies on how snowmobiles affect the snowpack. In fact, I also reviewed one of the only two studies cited in the manuscript [Thumlert and Jamieson, 2015] where the impacts of snowmobiles were quantitatively measured on a backcountry snowpack. Thus, there is a significant gap in the research, but the authors do not present convincing evidence that this gap is worth addressing. The authors need to motivate the study.
*>>The introduction has been rewritten to highlight the lack of research in this area, as well as the number of recreational users that this could impact.*

Why study changes in stratigraphy related to snowmobiles? Who will this research benefit?
*>>This work will benefit managers who need to make decisions about multi-use areas that are used by snowmobilers. As there has been limited related work, this also provides more quantitative information on how snowmobile use changes the snowpack. The text has been changed accordingly.*

The main conclusion that I came away with from this study is that regular snowmobile use, starting with a thin (30 cm) snowpack, results in a denser and harder snowpack with smaller basal grains. That conclusion is unsurprising, in that it could likely be predicted based on a basic understanding of snow mechanics, but given the lack of study on snowmobile effects, I still suggest the results are worth publishing. However, I worry that a reader might be tempted to conclude that snowmobiles can be used to strengthen the snowpack and prevent avalanches that fail on basal facets, similar to a boot packing program [e.g. Sahn, 2010]. While this may be true for isolated small areas, I cannot see backcountry snowmobile use reducing avalanche hazard, as the tracks will never carpet a slope densely enough. The authors should consider addressing this problematic conclusion that readers may come away with.
*>>This is an interesting comment. This has been added to the discussion.*

Sahn, K. (2010), Avalanche risk reduction in the continental climate: How to implement an effective boot packing program, Proceedings of the 2010 International Snow Science Workshop, p. 296-301.
Thumlert, S., and B. Jamieson (2015), Stress measurements from common snow slope stability tests, Cold Regions Science and Technology, 110, 38-46, [doi:10.1016/j.coldregions.2014.11.005].

Specific comments on the manuscript:
Line 25: These numbers have been removed in the rewrite
Line 27 (two comments): these are locations, and this has been removed
Line 30: I do not think that the specific numbers are relevant here
Line 35: ""of". Lots of careless errors here. Was this proof read?" *"of" was added. Much of this*

*text has been rewritten as per Reviewer 1.*

Line 48: "Why are these objectives important? What's the motivation? " *The text has been rewritten, and a sentence has been added at the end of the Introduction to highlight the relevance of this work*

Line 77: "can you provide some numbers here? Medium and high relative to where? " *These are based on observations by the authors, and USFS staff who helped with the fieldwork. A personal communication has been added.*

Line 101: "type of snowmobile; weight of snowmobile; and speed of snowmobile" *the following was added "driving a Skidoo brand snowmobile weighing about 300 kg with the rider (Figure 2d) at 10 km/h"*

Line 113: "You should note that your depth measurements are measured from the ground going up" *this is added, although it is the standard to measure snow depth from the ground up and thus assumed.*

Line 140: "maximum diameter I am assuming?" *The word "mean" has been added*

Line 143: "No, hardness is penetration resistance (Fierz et al. 2009, p 6). It usually measured in Newtons, which is g m s^-2. You should say something like "...in this study hardness is reported as force per unit area..."" *This has been changed/added*

Line 188: "I'd like to see the bulk density over time plotted or in a Table" *A figure has been added*

Line 243: "As with the bulk density, it would show your findings better if there were a plot of mean hardness over time or a table." *A figure has been added*

Line 248: "These are interesting findings, especially for snow stability" *No change*

Line 310: "fix" *this sentence has been deleted*

Line 317: "constantly?" *this sentence has been deleted*

Line 323: "I don't like this description. Meteorology doesn't drive snowpack metamorphism from the surface down. It's the movement of water vapor through the snowpack that drives metamorphism. For instance, for basal depth hoar formation, the vapor flux is from the ground towards the snow surface." *this sentence has been deleted*

Line 383: "so what? Observations of > 100 deg C m^-1 are not uncommon for a thin snowpack" *the citation and comment have been deleted.*

Line 383: "isothermal" *"al" has been added*

Line 392: "This belongs in the results" *this has been moved*

Line 437: "I am not convinced there's evidence from this study that snowmobile use increases SWE. In Section 4.5, you said the SWE was similar across all 3 sites." *What was mean was the mass of the snowmobile, not SWE. This sentence has been removed.*

Line 469: "experiments" *an "s' was added*

Figure 2: "perhaps, "8-Jun" ? The spacing on this axis is poorly chosen. 1st of the month for each month would be easier to follow or bimonthly" *This figure has been replaced.*

Figure 3: "Clarify in the caption whether depth is measured from the ground or the snow surface. It appears to be measured from the ground going up." *The sentence "the ground is at zero snow depth" has been added to the caption.*

[revised manuscript text omitted]

---

## Author Response (AR3)

> We appreciated the in-depth comments of Reviewer #1 and have rewritten much of the text, reordered the figures to present. We focus more on the time series of snowpack property change and have reduced the three layer-level detailed property plots to a single sample plot. We have added a simple densification model, as per the suggestion of the previous review of Reviewer #1. This model is calibrated on the experimental data, and evaluated on the operational dataset, with moderate success.

**REVIEWER #1**

The authors have made substantial changes to the original paper and addressed some of the concerns raised by the reviewers. While this has somewhat improved the quality of the work, the manuscript still requires major revisions to improve the presentation of the results and the discussion of their limitations and implications.

> As per this reviewer's previous comments, a bulk density change model has been created with the experimental data and evaluated on the operational data.

The introduction has in part been rewritten. Nevertheless, it still does not provide a clear context to explain why this study is requires and why one should be interested in the influence of snowmobile use on snow properties.

> We want to study the specifics of how snowmobile use impacts the snowpack, so the focus of the paper remains that. This is stated in the Introduction.

Does it affect the underlying vegetation, is it relevant for snow melt in the spring, does is stabilize the snow cover to reduce the avalanche danger?

> in the Introduction we state that the amount of snowmobile use is increasing in many locations and that will further change the snowpack. We further add some text related to the impacts on underlying vegetation and potential avalanche risk.

There is limited research on how snowmobile activity influences underlying vegetation (except Keddy et al., 1979), so the addition of snow due to snowmaking provides an indication of possible changes. It was found that there is often more soil frost, ice layers may form at the base of the snowpack, and there is often a delay in vegetative growth due to extended snow cover (Rixen et al., 2003). Model simulation have snow snowmelt can occur later due to a denser snowpack and more heat loss from the snowpack and underlaying soil (Fassnacht and Soulis, 2002); increased snow loading (Rixen et al., 2003) and manual compaction (Martz et al., 2016) yield cold soil.

Also, a changing climate could cause more compaction (Martz et al., 2016).

I suspect the last topic was what motivated the authors to perform these measurement. If so, this should clearly be stated in the introduction, and relevant studies which have investigated these effects should be discussed.

> We are not interested in how compaction can stabilize the snow cover to reduce the avalanche danger. In fact, we stated in the Discussion we caution against this method.

The presentation of the results has not improved much and still remains rather poor. Most figures

show vertical density, temperature, hardness and ramm hardness profiles for all sampling dates. > While we disagree about the detailed plots, we have reduced them to a single date to illustrate differences in layers or sampling intervals. We have added basal hardness and basal crystal/grain size and shape to the previous time series plot (Figure 4). It now shows the key results.

**These figures are illustrative but not easy to interpret.**

> The plots of density, hardness and ram resistance versus depth provide detailed differences between treatment methods. However, they have been reduce to one date (mid-February and mi-experiment) and the emphasis is now on the time series plot.

Furthermore, the authors mainly discuss mean (bulk) properties or the properties of the basal layer. While the authors have now included a Figure showing the evolution of mean density, basal density, mean temperature gradient and mean hardness (Figure 8), this figure is only briefly mentioned at the end of the results section in a separate subsection (4.6). > We have now incorporated a more detailed summary of these time series results, and de-emphasized the individual layer-based results in the other four figures. The time series figure is presented prior to the other figures. We now use it as the basis of presenting the results and the use the other four for backing up specific points.

Furthermore, many of the results shown in this figure are repeatedly discussed before. For instance, in lines 208 to 220 the authors discuss changes in bulk density and constantly refer to Figure 4, which shows the vertical density profiles. While reading this passage, I found myself repeatedly looking at Figure 8, and it would be much more efficient and intuitive for the reader to show the plots of the mean and basal properties in each respective subsection. > Figure 8 (time series) has been moved to before Figure 4 and the Results section has been rewritten using the time series as the starting point to illustrate overall differences with the other figures (previously 4 to 7) used to illustrate detailed differences. The Results is now a summary with less specifics on layers.

Finally, the discussion and conclusion sections still need to be largely rewritten as it remains very scattered. Indeed, the authors need to do a much better job at putting their results into context, discuss the limitations of their methodologies and findings and highlight new insights. For instance, the hardness and ramm measurements have some peculiarities. In some pits specific layers sometimes have very high values which then disappear in subsequent pits. This is not observed in the control pits and highlight the difficulties in obtaining reliable hardness measurements in snow disturbed by snowmobiles. Such problems are not discussed at all by the authors even though they clearly highlight some of the limitations of this study. > *The Discussion has been reorganized and various paragraphs have been combined. Much of the remaining text has been rewritten. A paragraph on Limitations has been added to the Discussion. The Conclusions have been rewritten.*

Similarly, the authors put a lot of weight on a 9 mm grain size measurement in one pit (section 4.5 and line 399 in the discussion) to discuss the influence of snow mobile travel on grain size. I have dug many snow pits and have looked at countless layers of depth hoar in various snow climates (from coastal to continental), and have seldom seen depth hoar crystals of that size. This

particular measurement is therefore rather surprising to me and could very well be an outlier, and the authors should be more cautious with their interpretation.

> The reviewer states that they have "seldom" seen depth hoar crystals of 9 mm in size, which implies that they have seen crystals this large. We have rewritten this point to emphasize that it is the difference in crystal size between the control and treatment that is relevant, not the actual size. We actually dug two control pits on that date and the size range was 8 to 10 mm for both pits.

Specific comments:

lines 37-38: it is not clear to me why I should be interested in changes in snow properties due to snowmobile travel. The context is missing. > *text has been added*

lines 47-48: 'had a highly significant effect' In what way did this effect manifest itself? *> The vegetation was compressed. This sentence has been changed.*

line 57-58: 'land managers need to make decisions'. What kind of decisions do they need to make that this study will help improve? > *This sentence has been changed to describe the decisions of which users use what areas.*

line 147: 'where the temperature gradient was linear' it is not very clear what the authors mean here. The temperature gradient between two temperature measurements is always 'linear'. > This has been reworded. A linear segment was used from the snow-soil interface to a distance below the snow surface where temperature increases. We have removed the temperature plots, but could add them back in if it helps clarify this method.

line 157: 'fresh' is not an official crystal form. 'Precipitation particles' should be used. *> changed*

line 168: 'for each stratigraphic layer'. I assume that for thin layers this was not possible. Please state the minimum layer thickness where these hardness measurements could be made. > *I think line 162 is meant. A sentence has been added "All layers thicker than 5 cm were identified due to the 5-cm diameter of the plate."*

line 175: what do the authors mean by 'relative hardness'? > *This has been removed*.

lines 195-197: I would say that even for REP the snow depth was somewhat below average. *> yes, this has been changed.*

Section 4.1: include a figure showing the temporal evolution of the mean and basal layer density over the season (from Figure 8), as most of the discussion centers around bulk and basal layer density and not the vertical profiles.

> This Figure (was 8, now  $\overline{4}$ ) has been moved earlier, and is cited much more.

Section 4.2: include a figure showing the temporal evolution of the temperature gradient and the

basal layer temperature.

> We are examining the temperature gradient and have removed the discussion of basal layer temperatures. The temperature gradient is a relative measure, and we feel it is more important than the basal temperature. The basal temperature varied little (-1 to 0C).

line 255: 'by April 26 (Figure 5b)': this figure only shows values for 26 March. > *This figure has been removed and the text has been changed*.

Section 4.3: include a figure showing the temporal evolution of the mean and basal layer hardness over the season.

> The time series of mean hardness was included in the last version. The temporal evolution of basal hardness has been added to the time series plot.

Section 4.4: include a figure showing the temporal evolution of the mean and basal layer ram resistance over the season.

> Less emphasis is put on the ram resistance.

Section 4.5: include a figure showing the temporal evolution of grain size over the season. This is much more illustrative than a table. It also more clearly shows that the 9 mm measurement is likely an outlier, and that the most marked differences in grain size were at the FEF site and for the high use site at REP (see figures below)

> This has been added with crystal shape and Table 1 has been removed.

Section 4.5: this section seems redundant as all these results were already addressed in the sections above.

> this section has been removed and the text is included in the individual sections.

line 331: 'were similar' in what way? Describe the similarities and differences more precisely. *> This sentence has been reworded.*

Section 5: The discussion requires extensive rewriting to more clearly discuss some of the limitations of the employed methodology, highlight the main findings and discuss the results in context with other studies.

> The objectives have been revisited.

lines 339-345 Here you provide a general statement on observed densification and compare it with results from another study. In lines 355-361 you again discuss the observed densification more quantitatively. Clearly, these two sections should be combined.

> Most of the first paragraph has been deleted and the remainder has been combined with other sentences.

lines 348-352: I don't think that compacting the snow with a snowmobile alters the snow microstructure, unless you are compacting new snow. What was the snow type when you first compacted the snow in December? Also, snow hardness is predominantly determined by density, and not grain characteristics.

> agreed. We have rewritten this based on what we saw: compaction of fresh snow and

fragmenting of faceted crystals. At REP there was new snow every day we sampled.

lines 352-354: 'such changes' unclear what this refers to. Be more specific. > [line 345] This sentence has been removed.

lines 361-365: unclear what the point is here. > *I think that this has been removed - I am not exactly sure what is being referenced here.*

lines 373-374: I don't agree with this statement. Your results show that for the FEF site there were very little differences between the amount of use as the densification and grain size changes were similar for low, medium and high use. For the REP site, on the other hand, the differences were more pronounced.

This is one of the main findings of your work which should be highlighted and discussed much more clearly.

> That is not what we found - see Figure 4. Specifically the influence was much more at FEF than REP and there were differences between the amount of use.

lines 384-386: provide an explanation why the effect of snowmobile travel is less for deeper snow covers. To me, this would mean that the initial impact of snowmobile travel, when the snow cover is still very shallow, is decisive. > *Most of this paragraph has been removed*.

lines 387-388: I do not believe that compaction impeded faceting. However, the resulting faceted snow is likely stronger (better bonded). Did you observe differences in grain type at the base of the snow cover?

> We observed fragmentation of faceted crystals at REF.

lines 390-393: it is unclear to me how less dense snow at the base of the control plots indicates that more metamorphism took place. You can still have kinetic growth in denser snow. > *This has been removed*.

lines 401-403: 'results may be transferable': what results do you mean? > In the previous reviewer, the reviewer asked how the results were transferable. This sentence has been rewritten.

lines 404-407: I do not follow your reasoning here. The results clearly show that there was no significant difference in temperature gradient. You can therefore not conclude that the vapor pressure gradients and depth hoar growth was slower since you did not measure those. All you can say is that the densification at the start led to a decrease in grain size throughout the season. > *This paragraph has been deleted. The sentence "densification at the start led to a decrease in grain size throughout the season" was added earlier in the Discussion.*

lines 408-424: The point of this section is not clear to me. Suggest rewriting. > *This paragraph describes the land management decision implications of snowmobile use on multi-use lands. This has been slightly rewritten and moved to the end.*

lines 425-431: This is the first time where a context for the measurements is given. This should also be mentioned in the introduction, as this seems the main reason why these measurements were performed.

> The Introduction has been partially rewritten to provide more context. This paragraph has been moved to earlier in the Discussion.

lines 440-453: this last section does not seem very relevant to me. > We feel that this paragraph helps explain some of the results that we saw. It has been moved and a new first sentence has been added to set its context.

Section 6: The conclusions have to be rewritten to better highlight the main findings and their implications.

> The Conclusions section has been rewritten.

> New citations added:

- Rixen, C., Stoeckli, V., and Ammann, W.: Does artificial snow production affect soil and vegetation of ski pistes? A review, Perspectives in Plant Ecology, Evolution and Systematics, 5(4), 219-230, 2003.
- Saly, D., Hendrikx, J., Birkeland, K., Challender, S., and Leonard, T.: The Effects of Compaction Methods on Snowpack Stability, Proceedings of the 2016 International Snow Science Workshop, Breckenridge, Colorado, 716-720, 2016.

- **Snowmobile Impacts on the**Snowpack Physical and Mechanical Properties of Different Steven R. Fassnacht1,2,3,4,5\*, Jared T. Heath1,65, Kelly J. Elder7Elder6, Niah B.H. Venable1,3 1
- 2
- 3 4
  - 1 Department of Ecosystem Science and Sustainability Watershed Science, Colorado State
- University, Fort Collins, Colorado USA 80523-1476 5
- 2 Cooperative Institute for Research in the Atmosphere, Fort Collins, Colorado USA 80523-1375 6
- 3 Geospatial Centroid at CSU, Fort Collins, Colorado USA 80523-1019 7
- 43 Natural Resources Ecology Laboratory, Fort Collins, Colorado USA 80523-1499 8
- 54 Geographisches Institut, Georg-August-Universität Göttingen, 37077 Göttingen, Germany 9
- 65 City of Fort Collins, Water Resources & Treatment, Fort Collins, Colorado USA 80521 10
- 76 Rocky Mountain Research Station, US Forest Service, Fort Collins, Colorado USA 80526 11
- \*Corresponding author: steven.fassnacht@colostate.edu; phone: +1.970.491.5454 12
- 13
- 14 Short title: Snowpack Changes due to Snowmobile Use

**15 Abstract**

| 16 | We ranSnowmobile use is a snowmobile over a seriespopular form of test plot towinter             |
|----|--------------------------------------------------------------------------------------------------|
| 17 | recreation in Colorado, particularly on public lands. To examine the physical and material       |
| 18 | properties effects of the differing levels of use on snowpack due to compaction from a           |
| 19 | snowmobile. We measured the snow density, temperature, stratigraphy, hardness, and ram           |
| 20 | resistance from snow pit profiles. Experimentsproperties, experiments were performed at two      |
| 21 | different experimental areas, specifically Rabbit Ears Pass near Steamboat Springs and at Fraser |
| 22 | Experimental Forest near Fraser, Colorado USA. We examined the differenceDifferences             |
| 23 | between no use and varying degrees of snowmobile use (low, medium and high) for different        |
| 24 | starts of snowmobile use, specifically on aon shallow (the operational standard of 30 cm) and    |
| 25 | deeper snowpacksnowpacks (120 cm). Significant changes in snowpack properties) were              |
| 26 | measured due to snowmobile use beginning on a shallow snowpack. These snowpackquantified         |
| 27 | and statistically assessed using measurements of snow density, temperature, stratigraphy,        |
| 28 | hardness, and ram resistance from snow pit profiles. Snowpack property changes were more         |
| 29 | pronounced where there was less snow accumulation. When snowmobile use started on ain            |
| 30 | deeper snow, in particular at 120cm conditions, there was less difference in density, hardness,  |
| 31 | and ram resistance compared to the control case of no snowmobile use. These results have         |
| 32 | implications for management of snowmobile use in times and places of shallower snow              |
| 33 | conditions where underlying natural resources could be affected by denser and harder             |
| 34 | snowpacks.                                                                                       |
| 35 |                                                                                                  |

**1. Introduction**

In the United States, where annually snowmobiling accounts for between \$7 billion 38 (American Council of Snowmobile Associations, 2014) to \$26 billion (International Snowmobile 39 Manufacturers Association, 2016) in annual revenue, and much of the snowmobile use isoccurs 40 on public land. The United States National Forest System seesrecords about 6 million annual 41 snowmobile visits annually, accessing about 327,000 km2 of land (US Forest Service, 2010 and 42 2013a). AsWith 
[revised manuscript text omitted]
|------------|----------------------------------|--------|----------------------------------|---------|--------|---------|
| a) Del     | a) Density                       |        |                                  | Low     | Medium | High    |
|            | Shallow initiation depth (30 cm) | Low    | < 0.01*                          |         |        | < 0.01* |
| DED        |                                  | High   | < 0.01*                          | < 0.01* |        |         |
| KEF        | Deep initiation depth (120 cm)   | Low    | 0.44                             | < 0.01* |        | < 0.01* |
|            |                                  | High   | 0.24                             | < 0.01* |        | < 0.01* |
|            |                                  | Low    | < 0.01*                          |         | 0.29   | 0.30    |
| FEF        | Shallow initiation depth (30 cm) | Medium | < 0.01*                          | 0.29    |        | 0.98    |
|            |                                  | High   | < 0.01*                          | 0.30    | 0.98   |         |

| 0 |                                                                     |                                      |        |      | Shallow initiation depth (30 |      |      |  |  |
|---|---------------------------------------------------------------------|--------------------------------------|--------|------|------------------------------|------|------|--|--|
|   | b) Temperature                                                      |                                      | No use | Low  | Medium                       | High |      |  |  |
|   | REP Shallow initiation depth (30 cm) Deep initiation depth (120 cm) | Low                                  | 0.22   |      |                              | 0.11 |      |  |  |
|   |                                                                     | Shahow initiation depth (50 cm)      | High   | 0.70 | 0.11                         |      |      |  |  |
|   |                                                                     | Deep initiation depth (120 cm)       | Low    | 0.77 | 0.34                         |      | 0.50 |  |  |
|   |                                                                     |                                      | High   | 1.00 | 0.22                         |      | 0.70 |  |  |
| c |                                                                     |                                      | Low    | 0.12 |                              | 0.89 | 0.10 |  |  |
|   | FEF                                                                 | FEF Shallow initiation depth (30 cm) | Medium | 0.14 | 0.89                         |      | 0.13 |  |  |
|   |                                                                     |                                      | High   | 0.64 | 0.10                         | 0.13 |      |  |  |
* * *
|    | a) Handmaaa |                                             |        |         | Shallow initiation depth (30 cm) |        |         |  |
|----|-------------|---------------------------------------------|--------|---------|----------------------------------|--------|---------|--|
|    | c) hardness |                                             |        | No use  | Low                              | Medium | High    |  |
|    | REP         | Shallow initiation depth (30 cm)            | Low    | < 0.01* |                                  |        | 0.16    |  |
|    |             |                                             | High   | < 0.01* | 0.16                             |        |         |  |
|    |             | Deep initiation depth (120 cm)              | Low    | 0.42    | < 0.01*                          |        | < 0.01* |  |
|    |             |                                             | High   | 0.06    | 0.02                             |        | < 0.01* |  |
| .0 |             |                                             | Low    | < 0.01* |                                  | 0.36   | 0.01    |  |
|    | FEF         | Shallow initiation depth (30 cm) Med
Hig | Medium | < 0.01* | 0.36                             |        | 0.08    |  |
|    |             |                                             | High   | < 0.01* | 0.01                             | 0.08   |         |  |

[revised manuscript text omitted]

---

## Author Response (AR4)

**Responses to Reviewers Comments:** Noted with a ">", with the response following.

The authors have made substantial changes to the original manuscript, and it now reads much better. In particular, the presentation of the results has improved, which now greatly facilitates the interpretation of the results. Nevertheless, I believe some changes are still required before the manuscript can be accepted for publication.

1. The presentation of the results obtained at the so-called operational sites can be improved. Specifically, I would suggest that the authors present the results shown in Figure 6 in a format similar to Figure 4. This would make the comparison of the experimental sites with the operational sites much more easy.
> *The plots of "point" measurements have been replaced by a time series of plots with the mean and basal values for the operational sites.*

2. Although the discussion section has improved, it is rather long and somewhat unstructured. The authors could use sub-sections with headers to provide a better structure and rearrange the paragraphs accordingly. I would suggest to split the discussion into three sub-section, e.g. (i) limitations of the measurement setup (ii) observed changes to snowpack properties and (iii) significance/impact of changes.
> *The discussion section has been significantly reorganized using the suggestions of the reviewer as a starting point and refining from there.*

3. The authors now provide a simple model to predict densification due to snow mobile usage based on the number of passes, snow depth and bulk density. This is a nice addition to the paper as it shows that changes in snowpack density could be modelled. However, in the current model, if the number of passes is 0, there can still be a change in density. Perhaps a model of the form:
$\Delta \rho_{s\_bulk} = A \times passes(B \times d_s + C \times \rho_{s\_bulk} + D)$
would be better suited, as the change in density would go to zero when there were no passes.
> *We used a non-linear version of the above model and were able to get a better fit. The fit to the operational sites was about the same.*

Finally, the authors should mention the potential merits of a density change model in the context of land use management in the discussion section
> *Mentioned model in discussion section.*

and their density change model should also be mentioned in the conclusions.
> *Completed.*

4. The authors should discuss the role of spatial variability in their results in a more quantitative manner. Indeed, the authors mention that they had two control transects at FEF (lines 126-127) and that the first measurements at FEF were performed prior to any snowmobile treatment (lines 138-140). Furthermore, the deeper snow treatments at REP only started on 1 February (lines 247-

249). All these data can be used to assess the typical degree of spatial variability at the experimental sites in a more quantitative manner.

> *We have added a figure illustrating the spatial variation in snowpack density, both mean and basal. This shows values at i) at REP deep snow (120 cm) compaction treatments (low and high use) on the first two sampling dates versus the control, ii) at FEF for the pre-treatment date (figure sampling date in Figures 5ii and 5iii) for the two sets of control snowpits at FEF. In the text we also mention a difference in hardness, but also acknowledge that the relative variation in hardness is larger than that of density since the non-treatment hardness values are so low (range of 0.4 to 5.8 kPa) versus the treatments (range of 30 to 1157 kPa).*

Detailed comments:

line 22: change 'where there was less snow accumulation' to 'for thinner snow accumulations'

> *Modified statement.*

line 48: on very shallow snow. Also, move the reference Keddy et al in line 50 to the end of line 48

> *Added "very" and copied the Keddy reference to line 48 but also left at end of 51.*

line 49: not clear how there can be an impact on the 'underlying old snow' if the snowpack is only 10 to 20 cm deep.

> *Removed/modified this reference as line 56 references vegetation.*

line 51: define what is meant by 'deeper snow cover'.

> *Added (>20 cm deep) to distinguish from the previous statement.*

line 60: not sure what is meant by 'greater heat loss from the snowpack and underlying soil'. Does this mean that there is more cooling in the snowpack and soil? This wouldn't make sense to me.

> *Modified sentence based on Fassnacht and Soulis, 2002 results to mean a delay in soil warming which can delay onset of vegetative growth.*

lines 68-69: I don't see how 'and billions of dollars are spent each year on snowmobiling' is relevant here and suggest removing it

> *Refined the statement.*

line 83: side of the pass …non-motorized users

> *Corrected and made a few edits to following sentences to improve readability.*

line 125: Two control transects: these are not shown in Fig. 2b.

> *Figure and statement modified.*

lines 154-155: remove this sentence

> *Modified sentence for clarity.*

line 167: not sure what the 'point of zero amplitude' is

*> Clarified, with new citation added.*

lines 172-173: remove 'each stratified layer of'
*> Modified sentence and removed.*

line 179: rewrite as 'It is due to' does not make sense to me. Hardness is not due to something, it is a property related to something
*> Reworded for clarity.*

line 184-186: Did you perform multiple measurements in thicker layers and average those values, or did you only perform one hardness measurement?
*> Clarified.*

line 191: replace 'tube' with 'rod'
*> Replaced.*

line 192: replace 'of known weight' with 'of defined weight'
*> Modified*

line 197: here you mention 'stratigraphic layers'. Are these the same as those identified in the manual snow profile? Usually, the layers identified in the ram profile do not correspond one-to-one to layers in the manual profile.
*> Clarified sampling. Stratigraphic layers are not the same as the depths used in ram sampling.*

line 204: The statement 'This determines the statistical significance between two datasets' is inaccurate. The Mann-Whitney test is used to compare two distributions and determine if these are statistically different from each other without assuming normal distributions.
*> Modified for clarity.*

lines 224-225: rewrite to: …REP had slightly below average snow depth compared to the 15 year mean based on the Columbine SNOTEL data…
*>Modified.*

line 226: rewrite to: …9 April was at 93% if the historical …
*> Done.*

line 229: rewrite to: …FEF was also below the 15 year …
*> Clarified.*

line 237: mention that this refers to the first data point in Figure 4ii
*> Adjusted.*

line 283: (Table 1c)
*> Corrected and added section references to Table 1 sections for all properties.*

line 296-297: remove sentence: 'These results are also …'

> *Done.*

line 307: move (Table 1) to end of sentence
> *Completed.*

line 317: fragmented facetted crystals is not an official crystal type mentioned in Fierz et al. (2009)
> *Adjusted statement to better reflect observations.*

lines 322 to 324: rewrite to …and snow depth (Figure 6a), the amount of snow was comparable for the … sites, even though they were up to …
> *Adjusted.*

line 326: the statement 'were similar' cannot be concluded based on what is shown in Fig. 6. Suggest showing the results as in Fig. 4
> *Revised for clarity, see new figure.*

lines 334-336: The line of reasoning does not make sense to me. Just because data do not fit the expected trend, does not mean they should be excluded. It is better to argument that you want to focus on dry snow conditions, and therefore exclude the data from later in the season.
> *Modified statements for clarity.*

lines 338-339: it is not clear to me what '…were not cross-correlated' means
> *Clarified statement.*

lines 345-349: Show the results which end up with a NSCE value of 0.71 in Fig. 7. Also, since you do not control the amount of snowmobile use at these sites, you can use the model to estimate it. It makes perfect sense that it varies throughout the season, as many factors influence the amount of use, including weather and time of year (holidays).
> *Revised section wording. Adjusted with new model.*

lines 354-357: Mention density changes in % to facilitate the comparison with literature values for grooming.
> *Calculated percentages for comparison.*

line 361: change 'densification' to 'density'
> *Done.*

line 362: 'compaction deformed fresh snow' not sure what this means and on what observations this statement is based.
> *Modified for clarity.*

lines 378-379: Figure 3ai and 3aii
> *This likely refers to a previous version of the document, Figure 3 only has an a) and b) now and the text reference should be correct.*

line 382: 'spatial variability between 40 to 200 kg/m3 for fresh snow' I strongly doubt that such a variability would be observed in the experimental sites. Clearly, some quantification of the spatial variability observed at the experimental sites would be in place to provide some context.
*> Clarified the statement, as a general observation based on previous studies and added more material based on new figure(s).*

lines 386-388: this sentence seems misplaced and should be moved to the paragraph in lines 409-415
*> Revised and moved.*

line 387: based on Figure 4, the crystals at the end of the season were no rounded crystals, but rounded facets.
*> Clarified.*

line 395: rewrite to: …property changes we observed could therefore also be …
*> Modified.*

line 411: …, it could impact weak layers that cause avalanches (Saly et al., 2016), which are typically soft layers consisting of large facetted grains (e.g. Schweizer and Jamieson, 2003; van Herwijnen and Jamieson, 2007)
*> Added and modified.*

line 415: 'Do not try …' rephrase this to say that the effects of snow mobile use on snow stability requires more investigation.
*> Adjusted.*

lines 428-430: also include Marty et al. (2017); Schmucki et al. (2015)
*> Added references.*

lines 432-440: move this paragraph to the start of the discussion
*> Changed when reorganizing the discussion section.*

*>Recommended references added:*

[revised manuscript text omitted]

**a) Rabbit Ears Pass sampling design**

meters low use (start at 30cm) | 2m
3m
high use (start at 120cm) | 2m
3m
control | 2m
3m
low use (start at 120cm) | 2m
3m
high use (start at 30cm) | 2m meters

**b) Fraser Experimental Forest sampling design**

meters low use | 2m
3m
medium use | 2m
3m
control | 2m
3m
high use | 2m meters c) pre-treatment d) during snowmobile treatment e) after treatment

[Figure]

**Figure 2.** The sampling design for the snow compaction plots at a) Rabbit Ears Pass, b) Fraser Experimental Forest, and photographs of the study plots c) pre-treatment, d) during treatment, and e) after treatment. The colors used for the control and treatment plots are used in Figures 45 through 78.

[Figure]

**Figure 3.** Mean snow depth from 2003-2017, and for the 2010 water year (WY2010) measured at a) the Columbine SNOTEL site near Rabbit Ears Pass (REP), Colorado and b) the Middle Fork Camp SNOTEL near Fraser Experimental Forest (FEF), Colorado, illustrating the dates of treatment and dates of sampling. Data were obtained online from the Natural Resource Conservation Service (NRCS) National Water and Climate Center (http://www.wcc.nrcs.usda.gov/).

[Figure]

[Figure]

[Figure]

**Figure 4**. Spatial variability of mean (yellow) and basal (blue) snowpack density by comparison of values at the Rabbit Ears Pass (REP shown with circles) deep snow (120 cm) compaction treatments (low and high use) and the control on the first two sampling dates, and at the Fraser Experiment Forest (FEF shown with triangles) for the two sets of control snowpits on the pre-treatment sampling date (see Figures 5i and 5ii, parts a) and b), respectively).

[Figure]

[Figure]

**Figure 5.** Time series for i. Rabbit Ear Pass (REP) and ii. Fraser Experimental Forest (FEF) at the different sampling dates of a) mean snowpack density, b) basal snowpack density, c) snowpack temperature gradient, d) mean snowpack hardness, e) basal layer hardness, and f) mean basal crystal size and shape. The crystal shape is included as per Fierz et al. (2009), with the exception of  faceted crystals that were fragmented. Note that the snowpack at the low and high use start at 30 cm could not be adequately tested for hardness on the first sampling date at the REP treatment plots.

[Figure]

**Figure 56.** a) Density, b) hardness, and c) ram resistance profiles for the February sampling dates (06 Feb at REP and 12 Feb at FEF) measured at the REP snow compaction study plot for no (control), low, and high use treatments beginning on i) 30 cm and ii) 120 cm of snow, and iii) the FEF snow compaction study plot for no (control), low, medium, and high use treatments beginning on 30 cm of snow. Note that free floating measurements represent overlapping density measurements. The ground is at zero snow depth.

[Figure]

[Figure]

**Figure 67.** Snowpit data for Walton Creek (no snowmobile use), Dumont Lakes (moderate snowmobile use) and Muddy Creek (high snowmobile use) in the Rabbit Ears Pass recreational use areas illustrating a) density, b) hardness, c) ram resistance, d) SWE, and e) snow depth. For a through c, the left panel (i) is the mean snowpack value and the right panel (ii) is the basal layer value.

[Figure]

**Figure 7̶8**. Bulk snowpack density change model for different amounts of use compared to the control of no use a) calibrated for the two experiment sites (Rabbit Ears Pass, REP and Fraser Experimental Forest, FEF), and b) applied to the operational sites (Dumont Lakes and Muddy Creek), compared to the no use Walton Creek site. The calibrated model is presented in a) with the Nash Sutcliffe Coefficient of Efficiency (NSCE). The NSCE is presented in b) for two different time periods̶.: the four pre-melt dates (December through March- 4 dates) and the later three pre-melt dates (January through March- JFM).

---

## Author Response (AR5)

The Editors comments were throughout the text and have been all been addressed with the tracked changes (and comments) version.

[revised manuscript text omitted]

**Comment [SRF1]:** this has been added

**Comment [SRF2]:** this has been modified

**Comment [SRF3]:** we added a sentence above that states that 2 or more samples were taken per 10 cm interval.

[revised manuscript text omitted]

> **Comment [SRF14]:** Once again, the description of Figure 4 is not sufficient to support this conclusion. Please consider expanding the discussion on spatial variability.
>
> > This description has been added to the start of the Results section.

[revised manuscript text omitted]

**Comment [SRF16]:** This whole caption is not very clear: what are the control and the comparison densities? Please clarify, and expand corresponding explanations in the text.
> changed

[Figure]

[Figure]

**Figure 5.** Time series for i. Rabbit Ear Pass (REP) and ii. Fraser Experimental Forest (FEF) at the different sampling dates of a) mean snowpack density, b) basal snowpack density, c) snowpack temperature gradient, d) mean snowpack hardness, e) basal layer hardness, and f) mean basal crystal size and shape. The crystal shape is included as per Fierz et al. (2009), with the exception of faceted crystals that were fragmented. Note that the snowpack at the low and high use start at 30 cm could not be adequately tested for hardness on the first sampling date at the REP treatment plots.

[Figure]

**Figure 6.** a) Density, b) hardness, and c) ram resistance profiles for the February sampling dates (06 Feb at REP and 12 Feb at FEF) measured at the REP snow compaction study plot for no (control), low, and high use treatments beginning on i) 30 cm and ii) 120 cm of snow, and iii) the FEF snow compaction study plot for no (control), low, medium, and high use treatments beginning on 30 cm of snow. Note that free floating measurements represent overlapping density measurements. The ground is at zero snow depth.

[Figure]

**Figure 7.** Snowpit data for Walton Creek (no snowmobile use), Dumont Lakes (moderate snowmobile use) and Muddy Creek (high snowmobile use) in the Rabbit Ears Pass recreational use areas illustrating a) density, b) hardness, c) ram resistance, d) SWE, and e) snow depth. For a through c, the left panel (i) is the mean snowpack value and the right panel (ii) is the basal layer value.

[Figure]

**Figure 8**. Bulk snowpack density change model for different amounts of use compared to the control of no use a) calibrated for the two experiment sites (Rabbit Ears Pass, REP and Fraser Experimental Forest, FEF), and b) applied to the operational sites (Dumont Lakes and Muddy Creek), compared to the no use Walton Creek site. The calibrated model is presented in a) with the Nash Sutcliffe Coefficient of Efficiency (NSCE). The NSCE is presented in b) for two different time periods: the four pre-melt dates (December through March- 4 dates) and the later three pre-melt dates (January through March- JFM).